# EAT: Entropy After $</\text{Think}>$ for reasoning model early exiting

## Abstract

Reasoning LLMs show improved performance with longer chains of thought. However, recent work has highlighted their tendency to overthink, continuing to revise answers even after reaching the correct solution. We quantitatively confirm this inefficiency from the distribution dynamics perspective by tracking Pass@1 for answers averaged over a large number of rollouts and find the model often begins to always produce the correct answer early in the reasoning, making extra reasoning tokens wasteful. To detect and prevent overthinking, we propose a simple and inexpensive novel signal, Entropy After $</\text{Think}>$ (`EAT`), for monitoring and deciding whether to exit reasoning early. By appending a stop thinking token (`</think>`) and monitoring the entropy of the following token as the model reasons, we obtain a trajectory that decreases and stabilizes when Pass@1 plateaus; thresholding its variance under an exponential moving average yields a practical stopping rule. Importantly, our approach enables adaptively allocating compute based on the `EAT` trajectory, allowing us to spend compute in a more efficient way compared with fixing the token budget for all questions. Empirically, on MATH-500 and AIME-2025, `EAT` reduces token usage by 12–22% without harming accuracy. `EAT` also remains effective in black-box settings where logits from the reasoning model are not accessible, and `EAT` is computed with proxy models: We verified the feasibility via early stopping Llama 70B with a 1.5B model and Claude 3.7 with a local 4B model.

## 1 Introduction

Large reasoning models such as GPT o1 (Jaech et al., 2024), DeepSeek R1 (DeepSeek-AI et al., 2025), or S1 (Muennighoff et al., 2025) exhibit the test-time scaling phe-

nomenon (Snell et al., 2024; Wu et al., 2024): when given more computation budget at inference, they generate longer chains of thought and achieve higher accuracy on reasoning tasks. This suggests that models can improve performance by "thinking longer," even without additional training.

However, current practice allocates a fixed token budget per question, regardless of difficulty. Easy questions can be solved confidently with just a few or no reasoning steps, while harder ones require extended deliberation. This uniform allocation wastes compute on simple instances while potentially underserving complex ones. This manifests as overthinking (Chen et al., 2024; Sui et al., 2025), where models unnecessarily revise correct reasoning, as illustrated in Appendix J. Quantitatively, we observe that Pass@1 accuracy (averaged over multiple random rollouts, Eq. (9)) often stabilizes and saturates early in the reasoning chain, sometimes even within the *first 10-20%* of the allocated token budget (Fig. 1, top row). This indicates that the model continues to generate the same answer regardless of additional reasoning steps, making further computation redundant.

This observation highlights the need for adaptive test-time computation mechanisms, where the model dynamically determines how much reasoning to allocate to each input. A natural principle for enabling such adaptivity is uncertainty: under submodularity assumptions, computation should continue as long as uncertainty decreases and stop once the uncertainty has stabilized. This idea has a long history in adaptive computing, from dynamically determining the number of steps in RNN (Graves, 2016) to confidence-based early exiting rules in deep networks (Teerapittayanon et al., 2016; Schuster et al., 2022). In the context of reasoning models, however, it is not clear whether there exist effective and inexpensive uncertainty signals for this purpose.

In this work, we propose Entropy After $</\text{Think}>$ (`EAT`) as such a signal (Sec. 4.1). By appending a stop thinking token (`</think>`) step and measuring the entropy of the model's next-token distribution over the single token after `</think>`, which captures the information gain of reasoning, we obtain a quantity that sharply decreases and stabilizes at precisely the point where accuracy saturates (Fig. 1, bottom row). When monitored over time with an exponential moving average, the variance of `EAT` provides a reliable criterion for early exiting (Alg. 1, Sec. 4.2).

---

[1]Anonymous Institution, Anonymous City, Anonymous Region, Anonymous Country. Correspondence to: Anonymous Author <anon.email@domain.com>.

Preliminary work. Under review by the International Conference on Machine Learning (ICML). Do not distribute.

In summary, this work makes the following contributions. **(i)** We provide the first quantitative demonstration that reasoning models overthink from a *distribution dynamics perspective*, motivating the usage of uncertainty as an early exit criterion (detailed arguments explained in Appendix B). **(ii)** We introduce Entropy After `</Think>` (EAT), a lightweight informative signal for early exiting, and propose a practical variance-based stopping rule using an exponential moving average. **(iii)** We empirically validate the effectiveness of EAT on benchmarks including MATH-500 and AIME-2025, showing that it reduces token usage by up to 21% without sacrificing accuracy (Sec. 5). Compared with existing approaches, EAT does not require sampling rollouts for computing the uncertainty signal and is applicable to black-box API models without logits access. **(iv)** To support future research and reproducibility, we also release large-scale answer rollouts and intermediate reasoning traces (taking over 20K GPU hours of computation), enabling future works to study early exiting without re-running costly experiments.

## 2 Related work

**Early exiting of deep learning models** In deep learning literature, it is common to perform layer-wise early exiting to prevent overthinking (Kaya et al., 2019) and improve efficiency. Using confidence / uncertainty as a metric for early exiting is a widely adopted wisdom. In particular, it has been shown that confidence scores can early exit CNNs, diffusion or LLMs (Graves, 2016; Teerapittayanon et al., 2016; Elbayad et al., 2019; Schuster et al., 2022; Jazbec et al., 2024). Our approach shares a similar spirit in that we also propose to use entropy, an uncertainty-related metric, as an early exiting criterion, but for reasoning process.

**Early exiting of long chain of thought** Since the release of open-source reasoning models, in particular, the DeepSeek R1 series, many recent works study early exiting. These methods share similar principles: they generate an answer rollout string during reasoning and look at various properties of the rollout related to uncertainty. Pioneering works such as Yang et al. (2025b) look at confidence score, measured by length-normalized log-likelihood (Malinin & Gales, 2020) of all tokens in the answer. Liu & Wang (2025); Fu et al. (2025); Mao et al. (2025) study whether the answer has stopped changing. Yong et al. (2025) look at the entropy empirically estimated through generating multiple rollouts using beam search. Generating a rollout is an expensive operation and is non-trivial in the sense that it often involves additional decoding hyperparameters and engineering techniques for efficiency. For example, Yang et al. (2025b) use branch-parallel decoding and only evaluate the metric when special keywords (such as "wait") pop up for efficiency. Liu & Wang (2025) perform a tree-based search over completions in order to estimate the entropy. In contrast, we demonstrate that the entropy over just one single token after

`</think>` is informative and rollout may not be necessary. This results in large efficiency gains. (Zhang et al., 2025) train a classifier on a labeled validation set to probe a reasoning model's hidden states to predict when to exit. EAT, in contrast, does not require *any training*, *labeled data*, or hidden state access, a stronger assumption than logit access. Of course, classifier based approach, as a *supervised approach*, can certainly give performance superior to EAT under ideal generalization. Lastly, unlike previous works, we demonstrate compatibility with black-box models: we can perform early exiting via only monitoring the verbal output of the reasoning model with a white-box proxy model without using the black-box reasoning model's internal information.

**Reasoning and entropy** Entropy plays an important role in understanding the reasoning process. Ton et al. (2025) study LLM reasoning from an information theory perspective; they made arguments similar to our work in that they suggest information gain / uncertainty should gradually increase / decrease throughout the reasoning process. However, the main difference is that they trained a separate LLM to estimate the answer given the reasoning chain and look at the entropy provided by this separate LLM, whereas we directly look at the entropy after `</think>` token using an out-of-the-box reasoning model without any fine-tuning / training. Wang et al. (2025) and Cheng et al. (2025) study token-level entropy inside reasoning chains to improve the post-training, in particular, they suggest applying policy gradient only on high entropy tokens and include entropy regularization in the objective to encourage reasoning process's exploration. Recently, it has been shown that information gain can also serve as a reward for reinforcement pre-training (Hatamizadeh et al., 2025), using a quantity similar to Eq. (6).

## 3 Background

### 3.1 Large language models

A decoder-only large language model (LLM) is a next token prediction model. Given a pre-trained LLM denoted as $\theta$, with a vocabulary of $|V|$ tokens, it takes in a context string `C` and produces a predictive distribution over the next token after `C`, denoted as

$$f(\texttt{C}; \theta) = \boldsymbol{p} \in \Delta^{|V|-1}, \tag{1}$$

where $\Delta^{|V|-1} = \left\{ \boldsymbol{p} \in \mathbb{R}^{|V|} \; \middle| \; \boldsymbol{p}_i \geq 0 \;\; \forall i, \;\; \sum_{i=1}^{|V|} \boldsymbol{p}_i = 1 \right\}$ denotes a $|V| - 1$ dimensional simplex. The entropy of this discrete distribution is,

$$\mathbb{H}(f(\texttt{C}; \theta)) = \sum_{i=1}^{|V|} -\boldsymbol{p}_i \log \boldsymbol{p}_i, \tag{2}$$

a strictly non-negative quantity that measures the uncertainty over the next token.

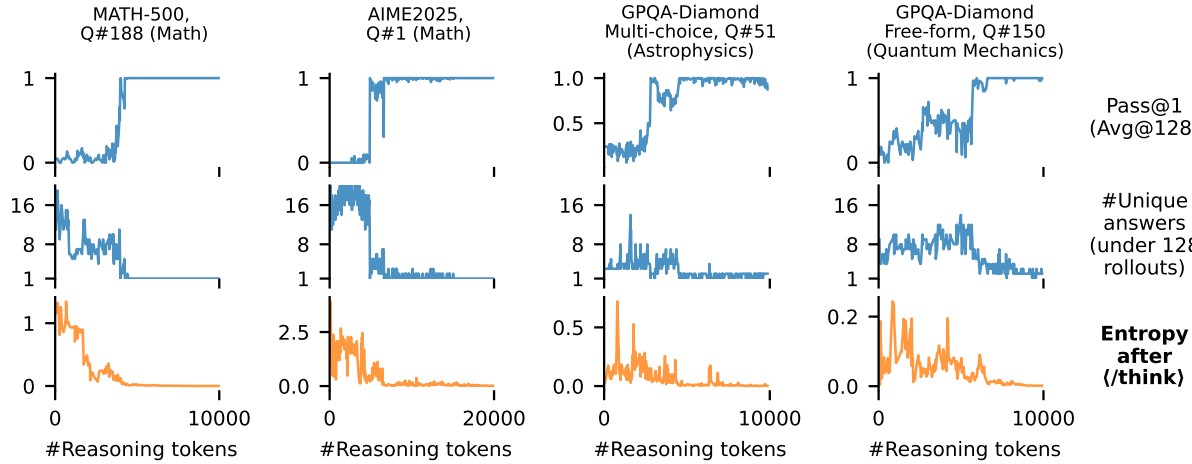

*Figure 1.* **EAT provides an informative signal to prevent overthinking in reasoning models.** We evaluate questions from four datasets (columns) using DeepSeek-R1-0528-Qwen3-8B, where we plot different metrics against the number of reasoning tokens. The first row shows that Pass@1 averaged over 128 rollouts (Eq. (9)) quickly saturates, indicating overthinking in reasoning. The number of unique answers under multiple rollouts (second row) stabilizes near one when Pass@1 converges; however, its evaluation has a high and non-deterministic overhead. We propose to manually append the stop thinking token (`</think>`) during reasoning and look at the **E**ntropy over the single token **A**fter `</T`hink> (EAT, bottom row, Eq. (5)), which drops and stabilizes at the point where Pass@1 plateaus, providing a cheap and deterministic signal for early exiting. For questions unsolved in the given budget, we present examples in Fig. 17, where EAT (as well as number of unique answers) does not show convergence or stablization and more reasons may be necessary.

Now given a context `C`, in order to generate a completion using an LLM, we iteratively sample the next token from $f(C; \theta)$ using certain strategies such as greedy decoding, random sampling, etc. after which the newly generated token will be added to `C` and we continue this iterative process until either a token budget is hit or some special tokens are generated. For the purpose of notation convenience, we define two special decoding operations

$$\mathrm{GenNewLine}(\cdot\,;\theta)\,;\mathrm{GenTillEoS}(\cdot\,;\theta) \qquad (3)$$

which denotes generating new tokens till a paragraph separation character `\n\n` is hit or End of Sequence token `<EoS>` is generated, respectively. By default, we apply random sampling for both operations unless a different decoding strategy is explicitly specified.

### 3.2 Reasoning models

Recently, it was shown that LLMs can perform reasoning after dedicated post-training (DeepSeek-AI et al., 2025). Reasoning models often generate very long chains of thought (more than 5K tokens), much longer than those generated by prompting the model to think step by step (Wei et al., 2022), often yielding a few hundred tokens.

Post-training encourages the outputs to be structured in a special format via reward design. To be more specific, it is often structured as

$$\mathrm{Q}, \texttt{<think>}, \underbrace{\mathrm{r_1}, \ldots, \mathrm{r_n}, \ldots, \mathrm{r_N}}_{\mathrm{R}}, \texttt{</think>}, \qquad (4)$$

$$\texttt{Final answer:}, \mathrm{A}$$

where commas denote the concatenation of the strings; `Q` and `A` denote the question and the generated answer, and `<think>` and `</think>` are special tokens denoting the beginning and end of thinking introduced during post-training. Between the begin / stop thinking token is the reasoning process `R`, where each `r_n` inside `R` denotes a line (ended with `\n\n`) in the reasoning process. If we read the reasoning lines verbally, several lines can compose a reasoning step, which constructs a complete step of self-revision, error-correction, etc. In practice, reasoning LLM shows the ability of "test time scaling": the more budget we allocate for `R`, the higher the chance of getting a correct answer.

### 3.3 Reasoning models overthink

One of the central goals of the post-training is to enable the LLM to generate longer responses to all questions. In fact, the response length is considered an important metric to monitor the progress of post-training, e.g. Fig. 3 in DeepSeek-AI et al. (2025), and in the model card of the latest DeepSeek release[1], the authors contrast it with an older release by how many more tokens it uses per question. Despite long chains of thought improving performance, for many questions, especially the easy ones, a big portion of the reasoning is redundant. The top row in Fig. 1 illustrates such overthinking behavior. After each reasoning line, we force the model to roll out an answer and measure the portion of correct answers under 128 randomly generated answer strings (Pass@1 Avg@128, Eq. (9)). We find that in the reasoning process, the metrics often stabilize at an

---

[1] https://huggingface.co/deepseek-ai/DeepSeek-R1-0528

early phase in the reasoning chain, long before the token budgets are used up or `</think>` is generated. Practically, this implies that there is no point in continuing to generate reasoning steps, as the model will be capable of getting a correct answer 100 percent of the time, at which point we can just stop thinking and elicit an answer.

Overthinking causes a high cost as the overhead of LLM outputs per token is much higher than that of the input tokens, due to the un-vectorizable and sequential nature of LLM decoding. It is also indicative of a broader problem, namely the need for more introspective capabilities for reasoning. This motivates us to develop methods for monitoring and early exiting the reasoning process, allowing us to halt the thinking process when the performance gain has saturated.

## 4  Method

In this section, we introduce Entropy After `</Think>` (EAT) in Sec. 4.1, a quantity to monitor the reasoning process and determine early exiting. We then discuss early exiting with EAT in Sec. 4.2 then efficiency considerations in Sec. 4.3

### 4.1  Entropy After `</think>`

Given a reasoning LLM $\theta$, we study the following quantity,

$$\texttt{EAT} = \mathbb{H}(f(\texttt{Q},\texttt{<think>}, \texttt{r}_1,\ldots,\texttt{r}_n,\texttt{</think>}; \theta)) \tag{5}$$

where $f(\cdot)$ and $\mathbb{H}(\cdot)$ are the next token prediction probability and entropy defined in Eq. (1) and Eq. (2). The motivation behind EAT is to study the **information gain** from the reasoning process. The information gain of a reasoning process $\texttt{R} = \texttt{r}_1,\ldots,\texttt{r}_n$ is the reduction in uncertainty, measured by the difference in entropy between the next-token distribution before and after reasoning,

$$\text{Information Gain} = \underbrace{\mathbb{H}\left(f\left(\texttt{Q},\texttt{<think>},\texttt{</think>};\theta\right)\right)}_{\text{Constant w.r.t reasoning steps}}$$
$$- \underbrace{\mathbb{H}\left(f\left(\texttt{Q},\texttt{<think>},\texttt{r}_1,\ldots,\texttt{r}_n,\texttt{</think>};\theta\right)\right)}_{\texttt{EAT}}. \tag{6}$$

Since the first term does not depend on the exact reasoning steps, EAT describes the information gain from R. It is worth emphasizing that EAT looks at only the entropy over one token; this differs from prior work that requires generating a full answer after appending `</think>`. Surprisingly, the entropy over just one token is highly informative: We visualize the examples EAT traces in the bottom row in Fig. 1. The EAT trajectory begins at a high value and gradually decreases and stabilizes, indicating that the single-token information gain gradually increases and then stabilizes. Importantly, when EAT stabilizes, the model's performance, measured by Pass@1 (Eq. (9)), also stabilizes. This correlation inspires our central idea to perform early

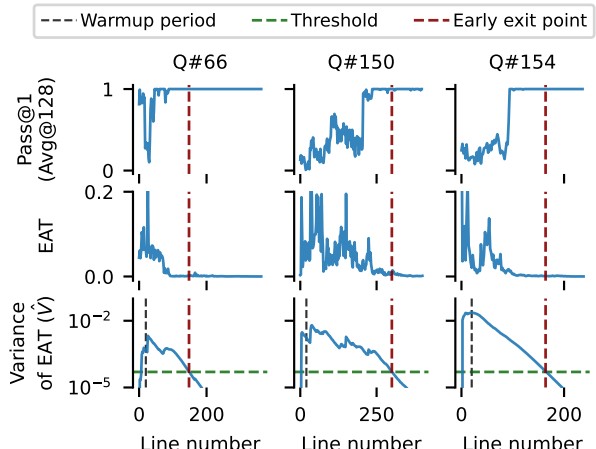

*Figure 2.* **Illustration of early exiting by thresholding the EMA estimated variance of `EAT`.** We evaluate DeepSeek0528-Qwen8B on various questions from free-form version of GPQA-Diamond (column title denotes question number). As reasoning proceeds, Pass@1 saturates, EAT stabilizes, and the variance of EAT ($\hat{V}$, Eq. (7), bottom row) decreases. Exiting the reasoning when $\hat{V}$ goes below the threshold (green line) avoids overthinking while maintaining high accuracy.

exiting of the reasoning chain with EAT when it stabilizes. Note that while we do not yet have a rigorous proof of this phenomenon, we provide an analysis in Appendix C that offers possible explanations. Lastly, EAT contrasts with the entropy inside the reasoning, which shows a much noisier pattern (Appendix F) and unclear correlation with Pass@1.

Additionally, we manually annotate a reasoning trajectory, marking the position where an answer is provided, and we consider these positions as separations of reasoning steps. We track the values of EAT at these positions (Fig. 7 , Appendix C): The trajectory at these positions shows a less noisy and more *monotonically decreasing pattern*, indicating each reasoning step reduces uncertainty.

### 4.2  Automatic early exiting

Fig. 1 shows that when EAT stabilizes, the performance gain from reasoning also saturates, which motivates us to develop an EAT-based criterion for automatic stopping. Alg. 1 shows the detailed end-to-end algorithm we propose. In particular, we look at the *exponential moving average* (EMA) estimation (Bruce, 1969) of EAT trajectory's variance. We maintain a running mean ($\hat{M}$) and variance ($\hat{V}$), and we update them every time a new reasoning line is generated, upon which a new value of EAT is computed:

$$\hat{M} = (1-\alpha)\hat{M} + \alpha\texttt{EAT}, \tag{7}$$
$$\hat{V} = (1-\alpha)\hat{V} + \alpha(\texttt{EAT} - \hat{M})^2, \tag{8}$$

where $\alpha \in (0,1)$ is the timescale parameter controlling the "window size" of the moving average. Empirically, we find that $\alpha \approx 0.2$ works well for most problems (Fig. 13, Appendix I.3). Intuitively, $\hat{V}$ approximately measures EAT's

variance over the past $1/\alpha$ iterations, and when $\hat{V}$ goes below a pre-defined threshold $\delta > 0$, we halt the reasoning.

The critical parameter in Alg. 1 is the threshold $\delta$, which allows one to *trade off compute for performance while preserving per-sample adaptivity*. Firstly, smaller values of $\delta$, i.e., a stricter stable condition, often take longer reasoning to achieve and vice versa. However, the actual reasoning budget is *adaptive*: For a fixed $\delta$, it takes less reasoning for EAT to stabilize and $\hat{V}$ to reach $\delta$ on easy problems, therefore fewer tokens will be used. On challenging problems, it takes more reasoning efforts, and Alg. 1 allocates more budgets. We provide examples of the early exiting procedure in Fig. 2, where we visualize traces of EAT, $\hat{V}$, and the early exiting point decided by the threshold $\delta$.

Importantly, EAT also applies in *black-box setting*, where only the reasoning model's verbal output is accessible, and its next-token probabilities are not. In such cases, we compute and monitor EAT with a proxy model, denoted as $\phi$ in Alg. 1. This helps in practice as large-scale reasoning models (e.g. $> 500B$ parameters) have to be deployed through a highly complicated pipeline spanned across multiple GPU nodes, where adding extra components can be non-trivial. The main reasoning model can remain massive, while the proxy can be a small 1.5B parameter model running on a single consumer GPU, yet still enable early exiting.

**Alternative evaluation frequency.** Note that despite Alg. 1 evaluates EAT every time "\n\n" is generated (line 4), EAT itself does not require any special structure in the reasoning to be effective[2]. We make this choice mainly for evaluation convenience and consistency with the accuracy metric: We compute Pass@1 (Eq. (9)) at every "\n\n", therefore we use the same schedule for EAT to enable direct assessment of early-stopping performance. In practice, other evaluation schedules are also feasible, e.g. every 100 tokens, we provide examples on EAT trajectory under alternative frequencies in Appendix G.

### 4.3 EAT is efficient to compute

Recall that our goal is to use EAT for early exiting in order for *efficiency*, therefore it is important to understand the overhead of evaluating EAT, because at the end of the day, the quantity we care about is not just the number of tokens in the reasoning chain, but the overall wall clock time.

The computation of EAT can be conducted very efficiently during the reasoning: Denote the size of the reasoning chain, measured by the number of tokens, as $|R|$. Since the KV cache of the reasoning chain is produced during decoding, computing EAT only requires one single forward pass, with a deterministic time and memory overhead linear with respect to $|R|$, roughly equivalent to the overhead of *generat-*

---

[2]However, we are unaware of any models that do not output "\n\n" in the reasoning.

---

**Algorithm 1** Early exiting reasoning process based on variance of EAT.

**Input:** Question $Q$, max token limit $T$; Variance threshold $\delta > 0$, EMA timescale $\alpha \in (0, 1)$.
**Input:** Reasoning LLM $\theta$; (*Optional*) A proxy LLM $\phi$ for computing EAT, if not provided, we let $\phi \leftarrow \theta$
**Output:** An answer string: A
1: Initialize EMA estimation of mean and variance $\hat{M}_0 \leftarrow 0$, $\hat{V}_0 \leftarrow 0$
2: Initialize reasoning line buffer $R \leftarrow []$ and line counter $n \leftarrow 0$
3: **while** $|R| < T$ **do**   ▷$|R|$ *denotes the total number of tokens*
4:    Generate new reasoning block
   $r_n \leftarrow \text{GenNewLine}(Q, \text{<think>}, R; \theta)$;
   *Or other conditions e.g., generate T extra tokens.*
5:    Append $r_n$ to R, update line counter $n \leftarrow n + 1$
6:    Compute
   $\text{EAT}_n \leftarrow \mathbb{H}(f(Q, \text{<think>}, R, \text{</think>}; \phi))$;  ▷*Eq. (5)*
7:    Update EMA $\hat{M}_n \leftarrow (1 - \alpha)\hat{M}_{n-1} + \alpha\text{EAT}_n$,
   $\hat{V}_n \leftarrow (1 - \alpha)\hat{V}_{n-1} + \alpha(\text{EAT}_n - \hat{M}_n)^2$    ▷*Eq. (7)*
8:    $\hat{V}'_n \leftarrow \frac{\hat{V}_n}{1-(1-\alpha)^n}$    ▷*De-biasing from the initialization*
9:    **if** ($\hat{V}'_n < \delta$) **or** `</think>` is generated **then**
10:      **Exit**
11: **Return** $A \sim \text{GenTillEos}(Q, \text{<think>}, R, \text{</think>}; \theta)$.

---

*ing one extra token*. We empirically verify the linear scaling of overhead in Fig. 6c, bottom row: Even with 8K tokens context, the overhead is under 0.1 seconds. This again contrasts EAT with rollout-based methods, which must generate multiple hypothetical continuations, introduce stochastic latency (since decoding time depends on the length of sequences), require non-trivial scheduling and parallelization to maximize efficiency, and often need pruning or beam search style optimization to keep compute manageable.

## 5 Experiments: Early exiting with EAT

In this section, we empirically evaluate EAT-based early exiting (Alg. 1). All experiments are mainly conducted on a GPU cluster with 40GB Nvidia A100 GPUs.

### 5.1 Experiment setup

**Models and datasets** We study four reasoning models: DeepSeek-R1-{0528-Qwen3-8B / Distill-Llama-70B}, Qwen3-4B-Thinking-2507, and Claude 3.7 Sonnet and three models for computing EAT: DeepSeek-R1-{Distill-Qwen-1.5B / 0528-Qwen3-8B}, and Qwen3-4B. The datasets used are Math500 (Lightman et al., 2023), AIME2025, and two versions of GPQA-Diamond (Rein et al., 2024): the standard *multiple-choice* version, where the model needs to pick one option from four candidates, and the *open-ended* version, where the model needs to generate its own answer. We also considered single-turn tool calling questions from the multiple-input subset of BFLC (Patil et al., 2025).

**Output format and correctness check** We prompt the model to put the final answer in `"\boxed{}"` following common practice. For Math500 and AIME2025, correctness

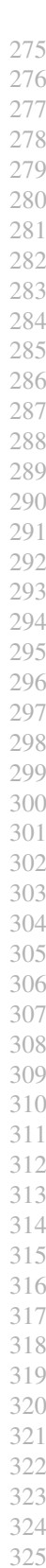
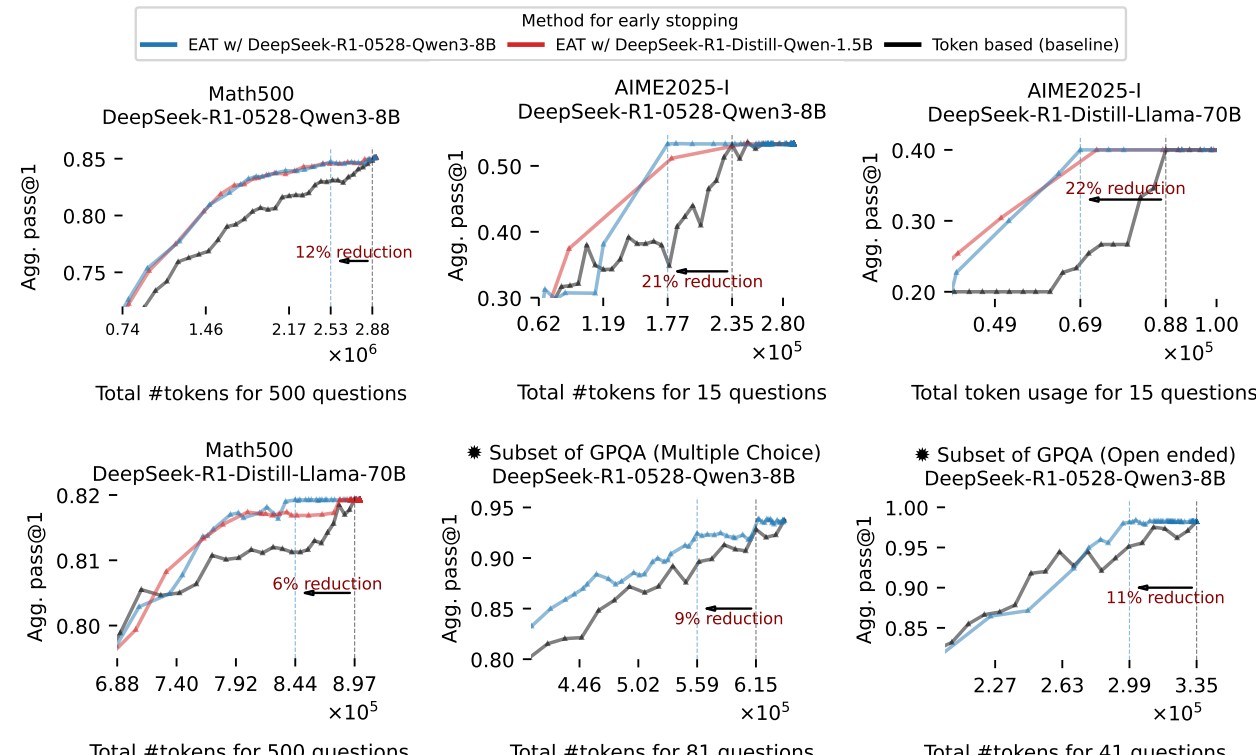

*Figure 3.* **EAT-based early exiting dynamically allocates token budgets and saves tokens without sacrificing accuracy.** Across different datasets and reasoning models (titles show dataset/model), thresholding the variance of `EAT` (blue and red lines, dot denotes a threshold $\delta$ used in Alg. 1) reduces token usage compared to token-based early exiting (black line, dot denotes a fixed per-question token limit $T$), thanks to its adaptivity. Crucially, `EAT` generalizes across model sizes: small proxy models can reliably early-stop much larger reasoning models (e.g., using a 1.5B model to early exit Llama-70B), making the method applicable to black-box APIs. Note that `EAT` shows less token saving on Llama-70B than Qwen3-8B because Qwen3-8B is a newer and stronger model, which uses more tokens per question, achieving higher Pass@1 but also overthinking more, leaving greater room for early exit gains.

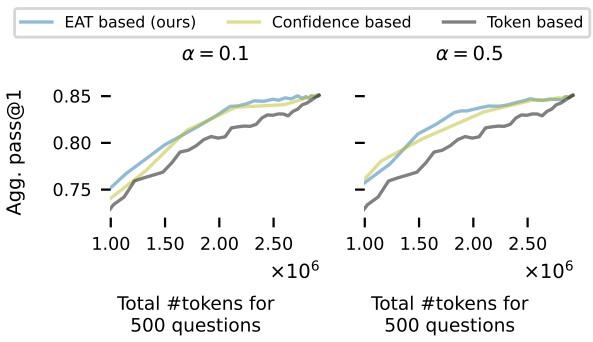

*Figure 4.* Evaluated on Math500 under DeepSeek Qwen3-8B, `EAT` shows similar early stopping performance compared with confidence (Eq. (16) in Appendix, Yang et al. (2025b)) computed by rolling out 5 tokens, at two different EMA windows sizes $\alpha$, while being 5 times cheaper for evaluation due to the rollout-free nature.

is checked by validating the equivalence of equations using SymPy, using the codebase released by Yang et al. (2024). For GPQA Multiple Choice, correctness is determined by exact match with the labeled option. For GPQA open-ended questions, correctness is judged using GPT-4o-mini with a verification prompt from Chandak et al. (2025). All prompts used in experiments are shown in Appendix K.

**Reasoning correctness metric** For each question in the dataset, with question as $Q_d$ and ground true answer as $A_d$, we track Pass@1(Avg@K) [3] after certain reasoning lines $R = r_1, \ldots, r_n$

$$\text{Pass@1}(\text{Avg@K})_d = \frac{1}{K} \sum_{k=1}^{K} \mathbb{1} \left[ A_d^k = A_d \right] \quad (9)$$

during the reasoning process, where $A_d^k$ denotes the $k$th randomly sampled rollout from

```
GenTillEoS(Q_d, <think>, R,
           </think>,Final answer:)  (10)
```

and the equal sign `"="` denotes whether $A_d^k$ is correct verified using techniques in the previous paragraph. Note that $A_d^k$ does not necessarily always start with `"\boxed{}"`, in practice, it sometimes starts with a summarization of the reasoning process followed by the actual answer. Additionally, the stop thinking token and the `"Final answer:"`

---

[3]Pass@1(Avg@K) is *never* used for determining when to early stop, it is only used as a metric for evaluating correctness.

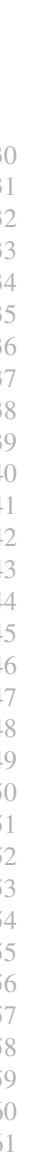
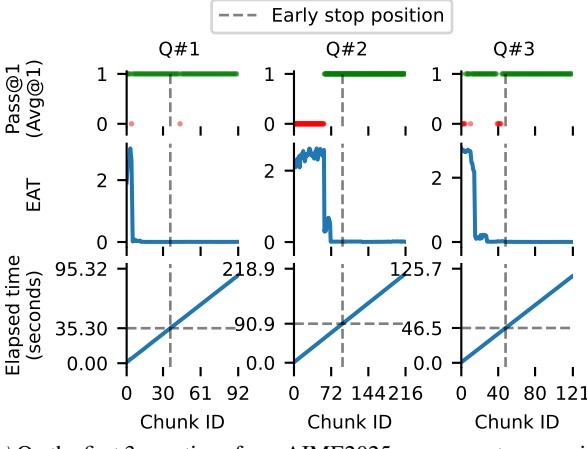

*(a)* On the first 3 questions from AIME2025, we generate reasoning using Claude 3.7 and compute `EAT` with Qwen3-4B-Thinking-2507. The gray dashed line denotes the early stopping position acquired by thresholding the variance at $10^{-3}$. `EAT` manages to save at least one minute for each query.

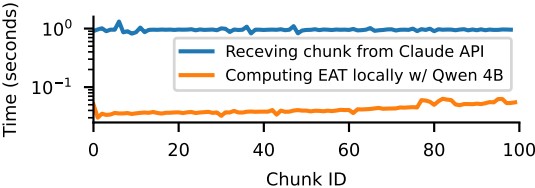

*(b)* Time for computing `EAT` on a local 4B model (computed on Nvidia L40s, averaged over 10 runs) is far smaller than the overhead of receiving a new chunk of reasoning text ($\sim 100$ tokens) from the Claude streaming API, enabling overlapped compute for black-box settings, introducing no wallclock overhead.

*Figure 5.* **`EAT` can early stop Claude 3.7 using a local Qwen 4B.** In case the logits of the API-based reasoning models are unavailable, but the reasoning trajectories are, `EAT` can be effectively applied through feeding the reasoning text into a local proxy model (Qwen 4B in this case), without inducing extra overhead.

strings are *manually added* to force the model to start generating an answer instead of continuing to think after `</think>`. This metric estimates how likely the model is to get the answer correct in one shot, a quantity critical in scenarios where verifiers are not accessible. At dataset level (of $D$ questions), we report the aggregated performance.

$$\text{Agg. pass@1} = \frac{1}{D} \sum_{d=1}^{D} \text{Pass@1(Avg@K)}_d. \quad (11)$$

**Implementation details** Details on decoding configurations, LLM inference framework, and efficient implementations of early exiting experiments are shown in Appendix H.

### 5.2 Baseline methods and efficiency metrics

We compare `EAT`-based early exiting with two other early exiting strategies

- **Token-based (Alg. 2)** We set a *fixed* reasoning budget, denoted as $T$, per question. The reasoning stops when either

condition is met: i) the stop thinking token `</think>` is generated; ii) the total reasoning length hits $T$. The advantage is its simplicity and clear physical meaning: The actual overhead is bounded by $\mathcal{O}(D \times T)$. Its drawback lies in the lack of adaptivity: A uniform allocation does not take into account the problem difficulty and the actual progress of the reasoning process.

- **Number of unique answers in K rollouts (#UA@K, Alg. 3)** We use the number of distinct answers in $K$ rollouts answer string as early exiting signal. The second row in Fig. 1 provides an illustration: As reasoning progresses, #UA@128 typically decreases and stabilizes at 1, aligning with Pass@1 convergence. Early exiting can be triggered when #UA@K is smaller than a threshold $\Delta \in \mathbb{N}^+$ or when a token limit $T$ is hit. The advantage of #UA@K lies in its simplicity and adaptivity, as it requires only verbal forms of rollouts, without any model internal information, and the actual token usage changes adaptively depending on the problem. The main drawback of #UA@K is its sensitivity to $K$'s value and the resulting performance-overhead trade-off: A small $K$ introduces too much noise to the signal, whereas a large $K$ causes too much extra overhead.

**Reasoning efficiency metric** Given a dataset, to quantify the goodness of different early exiting approaches, we enumerate over thresholds (the exact values and ranges depend on methods) to construct an Agg. pass@1 (Eq. (11)) v.s. actual total token usage curve. A more efficient early exiting approach should have a larger area under the curve, indicating that fewer tokens are required to achieve a target performance. Note that for all instances in a dataset, we use the same threshold; we believe adaptive thresholds (e.g., different $\delta$s) for each instance can further boost performance.

### 5.3 Results

Now we present the efficiency results of different methods. For `EAT` and #UA@K, we set the maximum token limit $T$ as $10K$ tokens, and for their thresholds, we sweep $\delta \in 2^{-\{0,1,\ldots,39\}}$ for `EAT` and $\Delta \in \{1, 2, 3\}$ and $K \in \{8, 16, 32\}$ for #UA@K; For token-based early exiting, we sweep the token limit $T \in 250 \times \{1, 2, \ldots, 40\}$. We include a prefix string "`\nFinal answer:`" (Eq. (13), Appendix D) for all models when computing `EAT`. The entropy is always computed over the full vocabulary logits.

**Comparison with token-based early exiting** Fig. 3 compares `EAT` with token-based early exiting (additional results on Qwen-4B are presented in Appendix I.1). Each dot on the lines denotes the Agg. pass@1 and actual total token usage under a specific threshold value, where dots on the black lines denote a value of $T$ for token-based early exiting, and dots on the red and blue lines denote a value of $\delta$ for `EAT` computed with two different models. For GPQA-Diamond, we filtered the dataset and only kept the problems

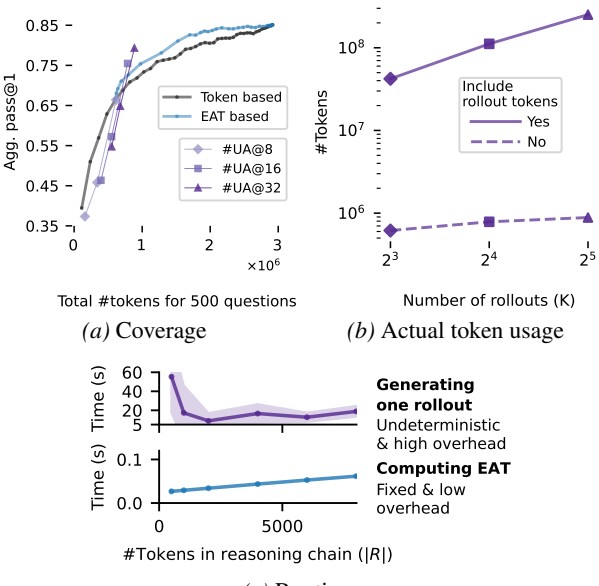

*(a)* Coverage      *(b)* Actual token usage

*(c)* Runtime

*Figure 6.* **#UA@K shows performance-overhead tradeoff** (a): #UA@K *only* works well when $K \geq 16$ (purple square line); (b): however, if we count the actual token (at $\Delta = 1$) required, including those from the $K$ rollouts, the number is very significant; (c): Generating rollout is expensive even for $K = 1$, and is more than 50 times slower than `EAT`. The runtime estimation of `EAT` includes the prefix string "`Final answer:`" and the rollout runtime is estimated with Huggingface implementation.

that the model solved eventually (more details discussed in Appendix I.4). Across datasets and models, `EAT` consistently reduces token usage (up to 21% on AIME-2025) while maintaining accuracy. Results for single-turn tool calling are shown in Fig. 12 in Appendix I.2: While we do still observe `EAT`'s convergence correlating with accuracy, it seems that reasoning is *not* necessary for this setting, therefore we do not claim `EAT` to be advantageous here.

**Comparison with confidence** We also compare `EAT` with the confidence score proposed by Yang et al. (2025b), where we estimate the confidence by greedily rolling out 5 tokens for the answer string and compute the length-normalized likelihood over them. The results are shown in Fig. 4, where we find that `EAT` shows similar performance compared with the confidence for early stopping, while being rollout-free.

**Black-box setting** `EAT` is effective in black-box setups, where the model for computing `EAT` differs from the underlying reasoning model. This allows us to use small proxy models (1.5B) to early exit Llama-70B without direct access to its logits (Fig. 3 bottom left, top right). Fig. 5 demonstrates using a local Qwen 4B model to early stop Claude 3.7 (full experiment setting and results shown in Appendix I.7), the only existing black-box model that exposes full reasoning trajectories through API. `EAT` remains effective in this case (Fig. 5a), and the computation can overlap with the API call latency (Fig. 5b), without introducing extra latency.

**Comparison with #UA@K** Fig. 6 compares `EAT` with #UA@K on DeepSeek0528-Qwen8B under Math500. Each purple line denotes a specific value of $K$ and each dot denotes a value of unique answer number thresholds $\Delta$. The metric value at each dot is the average over 64 random samples. The results show several critical issues with #UA@K. Firstly, the performance is sensitive to $K$: When $K$ is small (lightest diamond purple line in Fig. 6a), the estimation is inaccurate (since #UA@K is upper bounded by $K$) so #UA@K underestimates the actual value therefore stopping too early and underperforms other methods. When $K$ is large (darkest triangle purple line in Fig. 6a), we can achieve very good early exiting performance; however, the overhead will be very significant, which goes *against* the motivation of early exiting. The reason is that rollouts are expensive: generating them dominates both token usage (Fig. 6b) and runtime (Fig. 6c top) in a stochastic way.

**Error analysis of `EAT`** For GPQA, we only present early exiting results on solvable subsets in Fig. 3, where we only kept problems for which the models eventually reached a Pass@1 > 0.8. The reason is that there are many problems the model cannot solve and `EAT` will use up all $T$ tokens due to the $\hat{V}$ never going below $\delta$, we present examples and more discussions in Appendix I.4.

**Ablation study on `EAT`'s hyperparameter** We provide ablation studies on the effect of EMA timescale $\alpha$ and the effect of prefix string for early exiting performance in Appendix I.3, where we find that `EAT` remains effective as long as $\alpha > 0.1$, with or without the prefix string.

# 6 Limitation and future work

**`EAT` on unsolvable problems** On GPQA-Diamond, `EAT` often uses all token budget on unsolvable instances. We view this as a limitation of the "stop when confident/stable" paradigm, rather than a flaw of `EAT` as an uncertainty signal. A natural remedy is a complementary *lower-threshold* mechanism that halts when progress stalls, e.g., when `EAT`'s variance decays too slowly. A concurrent ICML 2026 submission demonstrates feasibility; we include an anonymized PDF in the supplementary material.

**Very long-horizon generation** `EAT` may be less informative for tasks with very long continuations after `</think>` (e.g., code generation), where the first token after `</think>` are weak proxy for downstream success. One option is to estimate signals from longer rollouts after `</think>`, which can be costly. Our black-box setting suggests a potential direction: Small reasoning models can effectively *monitor* larger ones' reasoning (also verified by a very recent work (Pal et al., 2026)). Therefore, one can consider more sophisticated signals from small models, e.g., prompting the small model with questions "Is the reasoning sufficient?" and checking the predictive probability.

## Impact Statement

This paper presents work whose goal is to advance the field of Machine Learning. There are many potential societal consequences of our work, none which we feel must be specifically highlighted here.

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

## A    LLM usage disclosure

Large language models are only moderately used for editing and improving the coherence of the text; they are not used to provide research or writing ideas in any form.

## B    Quantitative approaches for verifying overthinking

At the end of the introduction section (Sec. 1), we claim that we provide the first quantitative demonstration of overthinking from a distribution dynamic perspective. Here we provide a more detailed explanation on how our Pass@1 (Avg@128) metric differs from existing works' empirical evidence for overthinking:

To begin with, our argument is centered around the top two rows in Fig. 1, which aims at understanding the *dynamics* of the elicited predictive distribution throughout reasoning (i.e. $p(\text{answer} \mid \text{question, reasoning})$). Note that since the answer parts for free-form questions are usually of unknown length with unknown closed form distribution, the easiest way to understand their statistical properties is to independently *generate a large number of rollouts from this distribution* and study its empirical properties.

We believe our observation (top two rows of Fig. 1) tells us a more detailed picture of overthinking, in addition to just "The model can *solve* the question without using all reasoning tokens", in particular:

- In the beginning, the model is not capable of coming up with a correct answer. (Pass@1 $\approx 0$)
- After some reasoning, the model can answer correctly *by chance* ( Pass@1 $\in (0.1, 1)$)
- As reasoning proceeds, the model starts to always answer the question correctly (place for early stop), i.e. the model becomes fully confident in one single answer.

The observation indicates that reasoning gradually increases the model's chance of generating only a single answer, which is more often correct in one shot, i.e., *eliminating* the uncertainty in the predictive distribution, confirming the motivation for using confidence/entropy for early stopping.

Other related works analyze overthinking in complementary ways:

- Figure 1 of Yang et al. (2025b) look at a *single* rollout's correctness (Pass@1 (Avg@1)) throughout reasoning;
- Figure 2 in Fu et al. (2025) study the Pass@1 (Avg@1) at each step of multiple independently generated *reasoning chains* to verify "model can solve the question without using all tokens" is an observation consistent across independently sampled reasoning chains, while our perspective focuses the properties of independently generated answer at each step for one single reasoning chain.
- Figure 3 in Yong et al. (2025) shows that the model's uncertainty reduces throughout reasoning, but only in the multiple choice QA setting rather than free form generation problems.

## C  In-depth analysis of `EAT`

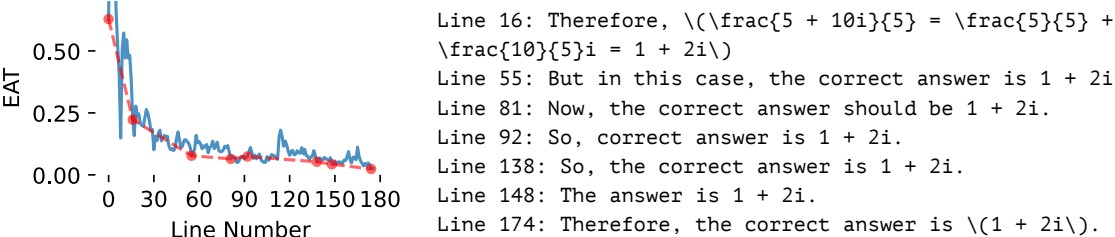

```
Line 16: Therefore, \(\frac{5 + 10i}{5} = \frac{5}{5} +
\frac{10}{5}i = 1 + 2i\)
Line 55: But in this case, the correct answer is 1 + 2i.
Line 81: Now, the correct answer should be 1 + 2i.
Line 92: So, correct answer is 1 + 2i.
Line 138: So, the correct answer is 1 + 2i.
Line 148: The answer is 1 + 2i.
Line 174: Therefore, the correct answer is \(1 + 2i\).
```

*Figure 7.* **EAT shows a monotonically decreasing pattern every time a conclusion is reached.** Intuitively, since `EAT` is related to information gain (Eq. (6)), we hypothesize that `EAT` will monotonically decrease at each reasoning step. However, in our experiments, since it is hard to know when a step has begun or ended, we evaluate `EAT` every line, and the `EAT` trajectory shows non-smooth patterns with lots of small bumps in the middle (blue line). However, if we look at the `EAT` values at each line where an answer is drawn (red dots, exact text shown on the right, manually annotated), which we consider as a "step", `EAT` trajectory shows a smoother decreasing pattern.

**Understanding the mechanisms behind `EAT`**  Assume the entire answer has N tokens, the N-gram entropy over an N-token autoregressive distribution, given a prompt $C$, is given by

$$\mathbb{H}_N(C) = \mathbb{E}_{T_1,\ldots,T_N|C}[-\log P(T_1,\ldots,T_N \mid C)]$$

where the expectation is taken with respect to all $|V|^N$ possible values ($|V|$ denotes the vocabulary size in the conditional distribution

$$P(T_1,...,T_N \mid C) = P(T_1 \mid C)P(T_2 \mid T_1,C),...,P(T_N \mid T_1,...,T_{N-1},C)$$

Importantly, we can decompose the N-gram entropy as

$$\mathbb{H}_N(C) = \mathbb{H}_1(C) + \mathbb{E}_{T_1|C}[\mathbb{H}_1(C,T_1)] + \mathbb{E}_{T_1,T_2|C}[\mathbb{H}_1(C,T_1,T_2)] + \ldots + \mathbb{E}_{T_1,T_2,\ldots,T_{N-1}|C}[\mathbb{H}_1(C,T_1,T_2,\ldots,T_{N-1})]$$

where $\mathbb{H}_1(\text{context})$ denotes the unigram entropy over the next token after `context`.

The decomposition is a summation of $N$ positive quantities, and the first term of the decomposition, $\mathbb{H}_1(C)$, corresponds to EAT. This tells us that

- When the uncertainty over the longer horizon, $\mathbb{H}_N(C)$, starts to shrink and stabilize as reasoning proceeds, e.g. if the perplexity / branching factor $\exp(\mathbb{H}_N(C)) \approx 1$, then `EAT` may also reduce and stabilize correspondingly.
- When EAT is at a high value and perturbs a lot, the longer horizon's uncertainty will also be high

However, note that our analysis does NOT answer the following three questions:

- Whether the first token's stabilization *leads to* the stabilization of the whole rollout answer. Note that this may depend on the model's linguistic pattern, where the first token may serve as the indicator for "High-level plan", similar to the role of branching token proposed in recent literature (Wang et al., 2025).
- Why does the uncertainty over the answer string reduce as reasoning proceeds? Some recent works observing this behavior (Yang & Holtzman, 2025; Li et al., 2025; Yang et al., 2025a), framing this as a "loss of diversity" issue, which can be problematic for e.g. creative writing. However, this behavior is also understandable in that post-training targets at maximizing Pass@1, which requires all rollout answer strings to contain one identical answer.
- Why EAT correlates with Pass@1. Our observation indicates that when the model is confident, it is often correct, as shown by the high Pass@1. This seems to imply that the model is often well calibrated, i.e. the uncertainty level is informative about the correctness. On the other hand, when the model shows low Pass@1, it often has a high and noisy EAT trajectory. It is currently unclear to us which component in the post-training pipeline provides the model with such a feature.

All three questions above are fundamental questions for understanding basic behavior of reasoning models beyond just `EAT`. While we think they are beyond the scope of the current manuscript, we will do better to explain the limitations of our understanding and connect it to these bigger questions and point to relevant literature where it exists. We do believe that our empirical observations, as well as the large-scale rollout we released, can motivate and accelerate future research along this line.

## D   Prefix string for computing `EAT`

In the main text, the `EAT` trajectories we illustrate in Fig. 1, 7 and 2 are computed as

$$\text{EAT} = \mathbb{H}(f(\text{Q}, \texttt{<think>}, \text{r}_1, \ldots, \text{r}_n, \texttt{</think>}, \backslash n; \theta)), \tag{12}$$

using the latest model DeepSeek0528-Qwen8B (released May 2025), where we observe that the stabilization of `EAT` correlates with the saturation of Pass@1.

For other models, such as DeepSeek-R1-Distill-Qwen-1.5B and DeepSeek-R1-Distill-Llama-8B (both released Feb 2025), we notice that a prefix string needs to be appended in order for `EAT` to be informative. In particular, we refer to the following quantity

$$\text{EAT}_{\text{prefix}} = \mathbb{H}(f(\text{Q}, \texttt{<think>}, \text{r}_1, \ldots, \text{r}_n, \texttt{</think>}, \backslash n, \texttt{The final answer:  }; \theta)), \tag{13}$$

which includes "The final answer:" followed by an empty space after the `</think>`\n.

We present some illustrations of `EAT` computed with and without the prefix under various models in Fig. 8a and 8b. Given reasoning trajectories from DeepSeek-R1-{0528-Qwen3-8B / Distill-Llama-70B}, we notice that $\text{EAT}_{\text{prefix}}$ is necessary for the old models (models released in late Jan), but not necessary for new models such as DeepSeek0528-Qwen8B and Qwen3-4B-Thinking-2507.

In practice, we find that using $\text{EAT}_{\text{prefix}}$ works better for early exiting performance, as it correlates more with Pass@1 compared with `EAT`, detailed results are presented in Appendix I.3. It is also worth noting that the memory and time overhead introduced by the extra prefix string is negligible in that all tokens inside the prefix string can be prefilled in parallel; therefore, the overhead will still be dominated by the length of the reasoning tokens.

## E   Early exiting algorithm

---

**Algorithm 2** Token-based early exiting

---

**Input:** Question Q, max token limit $T$;
**Input:** Reasoning LLM $\theta$
**Output:** An answer string: A
1: Initialize reasoning line buffer R $\leftarrow$ [] and line counter $n \leftarrow 0$.
2: **while** $|$R$| < T$ **do**                                         ▷$|R|$ *denotes the total number of tokens*
3:     Generate new reasoning line $\text{r}_n \leftarrow$ GenerateNewLine(Q, `<think>`, R; $\theta$);                    ▷*Eq. (3)*
4:     Append $\text{r}_n$ to R, update line counter $n \leftarrow n + 1$
5:     **if** `</think>` is generated **then**
6:         **Exit**
7: **Return** A $\sim$ GenTillEos(Q, `<think>`, R, `</think>`; $\theta$).

---

---

**Algorithm 3** #UA@$K$ based early exiting

---

**Input:** Question Q, token limit $T$; Number of rollouts $K$, unique answer threshold $\Delta$
**Input:** Reasoning LLM $\theta$
**Output:** An answer string: A
1: Initialize reasoning line buffer R $\leftarrow$ [] and line counter $n \leftarrow 0$.
2: **while** $|$R$| < T$ **do**                                         ▷$|R|$ *denotes the total number of tokens*
3:     Generate new reasoning line $\text{r}_n \leftarrow$ GenerateNewLine(Q, `<think>`, R; $\theta$);                    ▷*Eq. (3)*
4:     Append $\text{r}_n$ to R, update line counter $n \leftarrow n + 1$
5:     Randomly generate $K$ answer rollouts
     $\text{A}^k \sim$ GenTillEoS(Q, `<think>`, R, `</think>`, `Final answer:`\n; $\theta$), $k \in 1, \ldots, K$
6:     Extract and *count* all unique answers from $\text{A}^1, \ldots, \text{A}^K$, denoted as $U$
7:     **if** $(U \leq \Delta)$ **or** `</think>` is generated **then**
8:         **Exit**
9: **Return** A $\sim$ GenTillEos(Q, `<think>`, R, `</think>`; $\theta$).

---

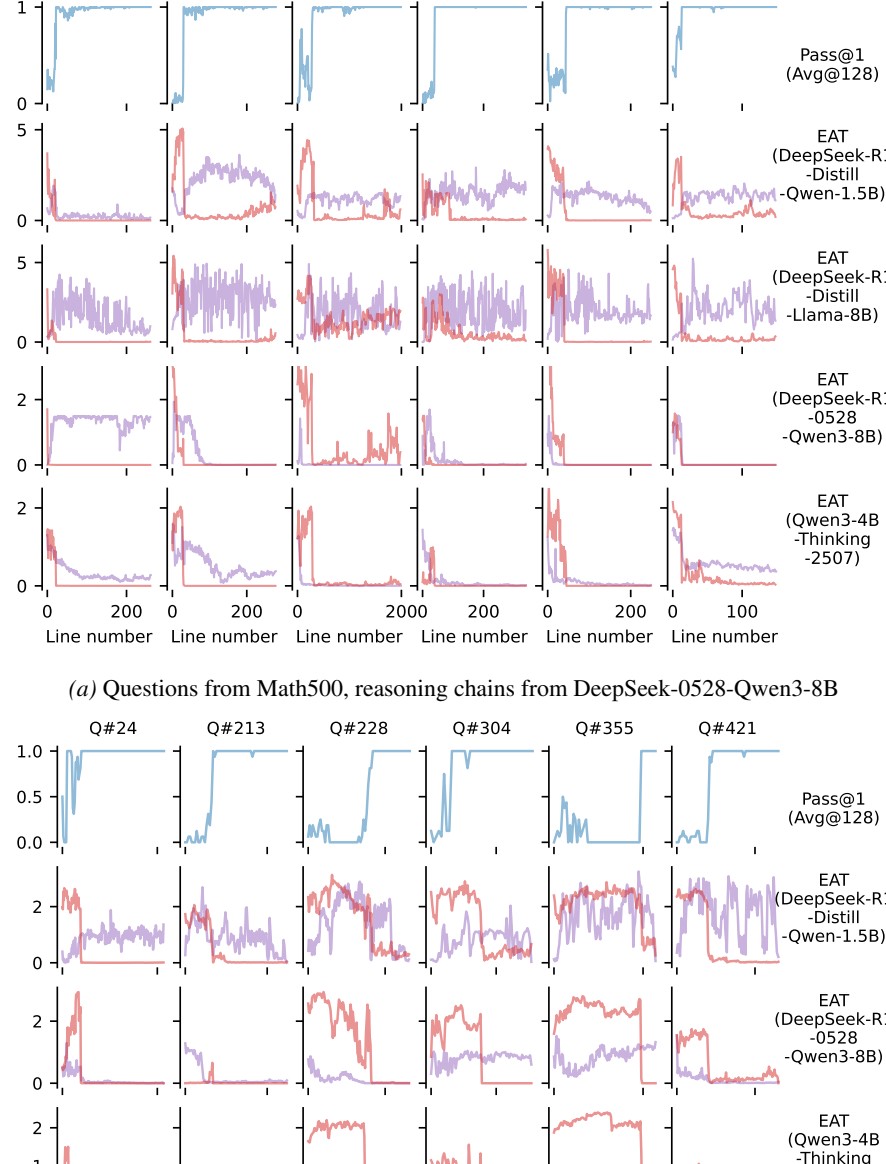

*(a)* Questions from Math500, reasoning chains from DeepSeek-0528-Qwen3-8B

*(b)* Questions from Math500, reasoning chains from DeepSeek-R1-Distill-Llama-70B

*Figure 8.* **EAT trajectories computed with and without a prefix string "The final answer:" for different reasoning models and Math500 questions.** )(a) Reasoning chains from DeepSeek-0528-Qwen3-8B: older proxy models (DeepSeek-R1-Distill-Qwen-1.5B and DeepSeek-R1-Distill-Llama-8B) require adding the prefix (red lines) for EAT to align with Pass@1 saturation, whereas the newer DeepSeek-0528-Qwen3-8B and Qwen3-4B-Thinking-2507 do not (purple lines). (b) Reasoning chains from DeepSeek-R1-Distill-Llama-70B show the same trend. In all cases, adding the prefix introduces negligible overhead because its tokens can be prefilled in parallel.

## F   Comparison with entropy after new line

Our empirical evidence shows that the entropy after `</think>` correlates with Pass@1 and the predictive distribution after. We did not find the entropy inside the chain of thought to be informative. In particular, we look at

$$\text{Entropy after newline} = \mathbb{H}(f(\mathbb{Q}, \texttt{<think>}, r_1, \ldots, r_n, \texttt{\textbackslash n\textbackslash n}; \theta)). \tag{14}$$

Examples are shown in the last row in Fig. 9.

We suspect that the entropy after newline may correlate with the internal stage of reasoning, e.g. whether the model (Wang et al., 2025) is in the middle of reasoning step or is about to finish and start a new round of revision, but less correlated with whether a confident answer is conditioned on the reasoning, a quantity Pass@1 reflects.

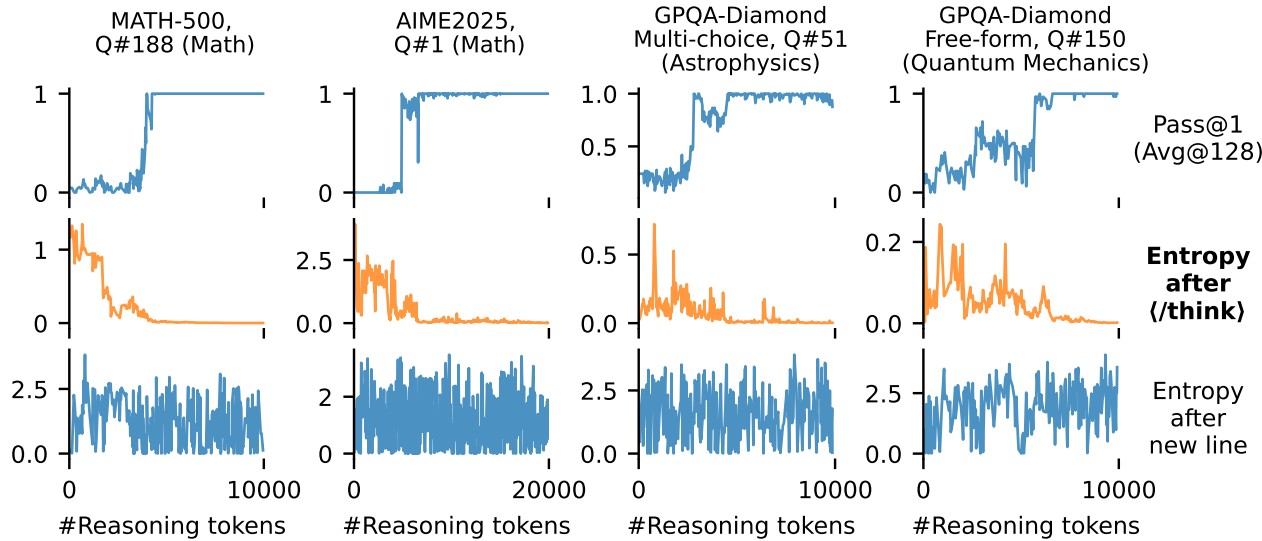

*Figure 9.* Similar to Fig. 1, but we include Entropy after newline (last row), which has the same overhead as `EAT` but is less informative and correlated with Pass@1.

## G   Evaluating EAT at alternative frequencies

In the main text, we compute EAT every time "\n\n" is generated; the main reason is that for the reasoning models we considered, the lengths distribute fairly equally across different paragraphs and we choose to compute the expensive Pass@1 (Avg@128) at every "\n\n" for implementation convenience. However, one may raise concerns that not all reasoning models output "\n\n", certain reasoning models could generate the whole CoT in one single long sentence (although it is unclear which model behaves like this).

Under such scenarios, one can also evaluate EAT at alternative scheduling, e.g., at every $S$-token. We provide illustrations of EAT trajectory evaluated every $S$-token under $S \in \{50, 100, 200\}$ in Fig. 10, where we find that the overall pattern of EAT stays unchanged, and EAT still correlates with the Pass@1 trajectory. However, it is *not* feasible to *evaluate* the actual early exiting performance under token-based scheduling as each dot on the EAT line no longer aligns with the points on the Pass@1 trajectory.

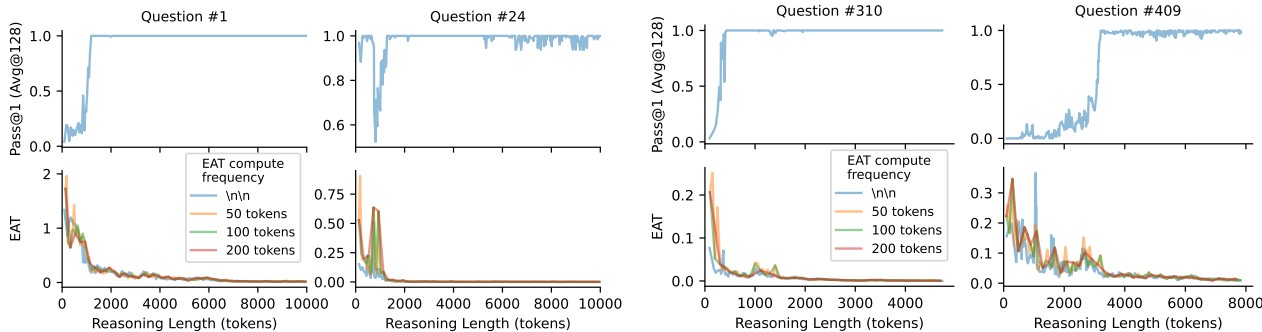

*Figure 10.* **EAT computed under different frequencies shows patterns.** Here we evaluate DeepSeek-R1-0528-Qwen3-8B on questions from Math500. Given the same reasoning trajectory, we evaluate EAT at different frequencies: Every new paragraph (blue line) and every $S$ tokens (the rest of the lines). The overall behavior of EAT stays unchanged, except that the trajectory becomes smoother at a higher value of $S$.

# H  Additional experiment setting

**Implementation details**  In our experiments, we generate the reasoning chain and compute `EAT` using the native Huggingface implementation *over the logits of the full vocabulary*. When generating answer rollouts, i.e. the one used to compute Eq. (9) and for estimating $\#UA@K$, we use vLLM (Kwon et al., 2023) for efficiency. For the generation of both the reasoning chains and the rollouts, we use a decoding configuration with temperature 0.6 and top-p of 0.95, following the official recommendations in the model card [4]. Due to resource limitations, for each question in each dataset, only one random reasoning chain is generated for evaluating early exiting.

**Simulated early exiting**  When conducting experiments, we perform early exit evaluation in a post-hoc manner. We first generate a single, very long chain of thought for each question, storing the complete reasoning trace and computing the Pass@1 and `EAT` trace. We then simulate early exiting by *retrospectively* truncating this chain at different stopping points. As such, for every instance in the dataset, we can analyze test time scaling and early exit behavior simply by producing one long reasoning chain, saving it once to disk, and replaying it offline to compute metrics at arbitrary exit thresholds without re-querying the model.

# I  Additional experiment results

## I.1  Experiment results on Qwen4B

Fig. 11 presents comparisons between `EAT`-based vs. token-based early exiting using Qwen3-4B-Thinking-2507 as the reasoning model evaluated on Math500 and AIME2025, and we compute `EAT` using three different models: Qwen3-4B-Thinking-2507 (green lines), DeepSeek-R1-Distill-Qwen-1.5B (red lines) and DeepSeek-R1-0528-Qwen3-8B (blue lines). The observation aligns with that in Fig. 3 in the main text, where `EAT` shows more efficient usage of tokens and remains effective when the model computing `EAT` differs from the reasoning model.

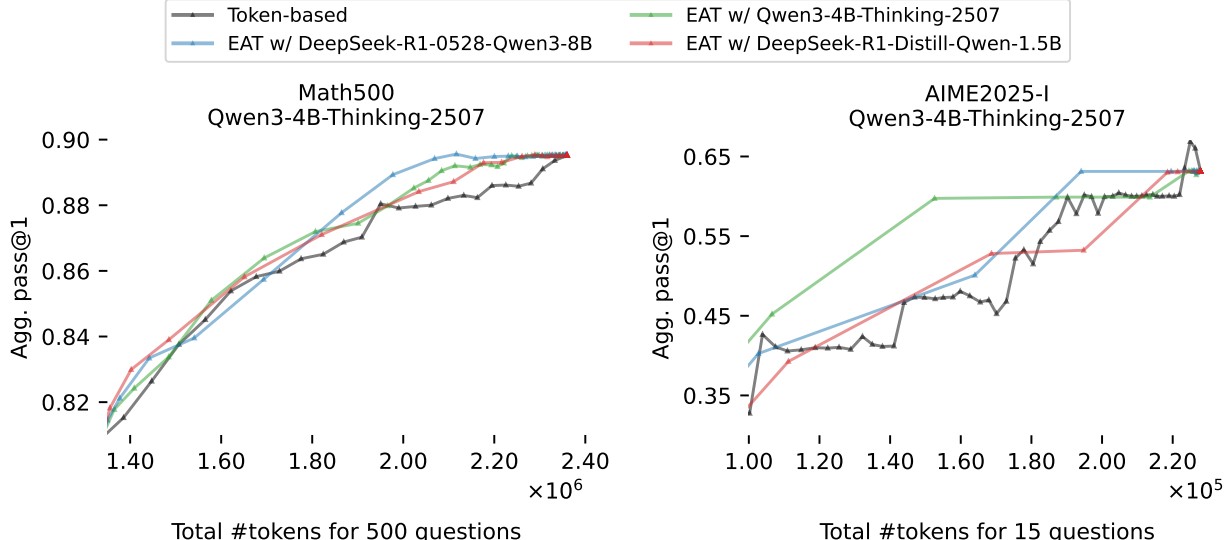

*Figure 11.* Performance of `EAT`-based early stopping with Qwen3-4B-Thinking-2507 as the reasoning model, experiment setting similar to Fig. 3 in the main text. Each dot on the black line denotes a per-question token limit threshold $T$ and each dot on the other lines denotes a variance threshold $\delta$. `EAT` saves tokens due to its adaptivity. Additionally, one can perform `EAT`-based early exiting using either the original reasoning model (green line) or other reasoning models (blue and red lines).

---

[4] https://huggingface.co/deepseek-ai/DeepSeek-R1-0528-Qwen3-8B
https://huggingface.co/deepseek-ai/DeepSeek-R1-Distill-Llama-70B

### I.2 Experiment results on tool calling

We considered single-turn, multi-input function questions from Berkeley Function Calling Leaderboard (Patil et al., 2025). For each question, we additionally include available tools from 5 other questions as a distractor. We use DeepSeek-Qwen3-8B, under the following prompt

```
You are an expert in composing functions. You are given a question and a set of possible functions.
Based on the question, you will need to make one or more function/tool calls to achieve the purpose.
Think step by step.
If none of the functions can be used, point it out. If the given question lacks the parameters required by the function, also point it out. You should only return the function call in tools call sections.
If you decide to invoke any of the function(s), you MUST put it in the format of [
func_name1(params_name1=params_value1, params_name2=params_value2...), func_name2(params)]
You SHOULD NOT include any other text in the response.
Here is a list of functions in JSON format that you can invoke.
```

where we compute `EAT` through appending `"["` after `</think>`

$$\text{EAT}_{\text{tool\_calling}} = \mathbb{H}(f(\texttt{Q}, \texttt{<think>}, r_1, \ldots, r_n, \texttt{</think>}, [; \theta)) \tag{15}$$

However, when evaluating the model outputs via Abstract Syntax Tree (AST) checker, we find that the majority of the error arises from formatting errors caused by the model failing to follow instructions (e.g., starting the response with the string `func_name=`) or including minor syntax errors (e.g., not wrapping a date in quotation marks as a string) that cause the AST parser to raise an exception. Through reasoning, sometimes the formatting error is corrected, sometimes it gets worse.

Additionally, for the majority of the questions, we find that the model manages to generate the correct tool call very early on in the reasoning, and we did not observe obvious evidence of test time scaling.

Now, assuming that the formatting error can be considered as an actual error, and the user insists to generate long chain of thought, we plot examples of Pass@1 (Avg@16) v.s. `EAT` in Fig. 12, where we manage to observe some relationship between the stability of `EAT` and the correctness. Regardless, it is overall unclear the value of a long chain of thoughts (or the value of `EAT`) in this single-turn tool calling setting, since one should turn off thinking mode for such a setting. We are also unable to claim that the adaptive budget allocation property of `EAT` can bring efficiency gain in this case.

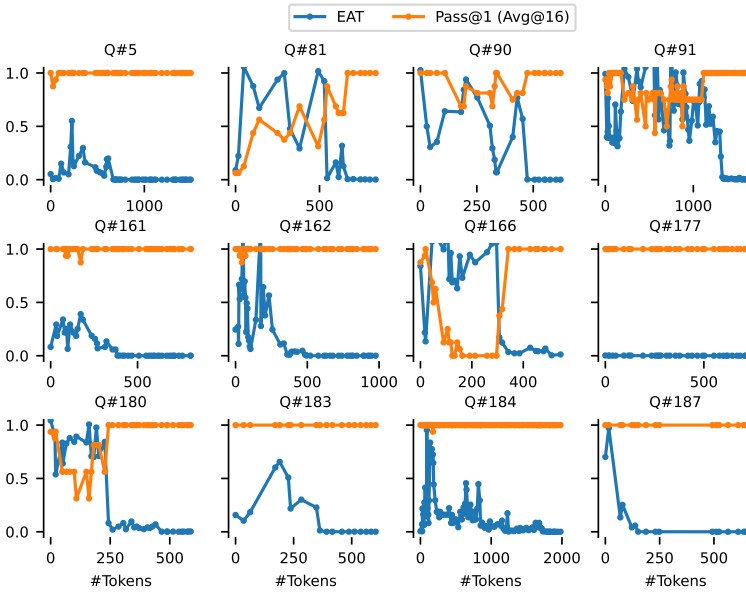

*Figure 12.* On single-turn tool calling benchmark problems, `EAT` serves as an informative signal for early stopping reasoning. However, reasoning overall does not seem to be necessary, as indicated by the high Pass@1 early in the reasoning.

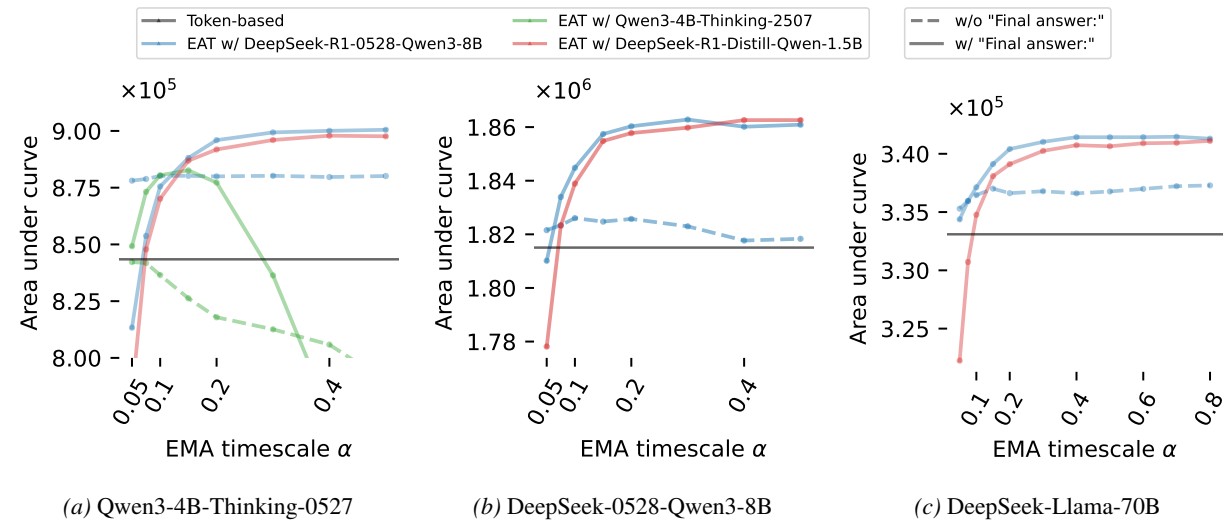

*(a)* Qwen3-4B-Thinking-0527        *(b)* DeepSeek-0528-Qwen3-8B        *(c)* DeepSeek-Llama-70B

*Figure 13.* **EAT outperforms token-based early stopping with or without prefix string included, under various values of EMA timescale $\alpha$.** Evaluated by the area under Agg. pass@1 v.s. total token usage curve (y-axis), where a larger value implies more efficient usage of tokens, we study three different reasoning models' performance on Math500 (subfigures), under various configurations of computing EAT (different lines) and values of timescale $\alpha$ (x-axis). Broadly, we find that a short time window, i.e. $\alpha$ value of about 0.1-0.2, works well for most methods.

### I.3   Ablation study

We conduct ablation studies to understand how the exponential moving average (EMA) timescale $\alpha$ and the influence of the optional `"Final answer:    "` prefix on the performance of EAT-based early exiting (also discussed in Appendix D).

For a fixed reasoning dataset (Math500), we sweep $\alpha$ over $\{0.01, 0.05, 0.1, 0.2, 0.4\}$. For each $\alpha$, we measure the area under the Pass@1–token-usage curve (AUC), which quantifies how efficiently a method maintains accuracy while saving tokens. We repeat the procedure both with and without appending the "Final answer: " prefix string when computing EAT.

Fig. 13 reports the resulting AUC as a function of $\alpha$ for Qwen3-4B-Thinking-2507, DeepSeek-R1-0528-Qwen3-8B and DeepSeek-R1-Distill-Llama-70B. EAT remains effective over a broad range of $\alpha$ values as long as $\alpha > 0.1$. Extremely small $\alpha$ (e.g., 0.01) averages over too long a window, blurring short-term fluctuations and causing delayed stopping, which lowers the AUC. In contrast, moderate $\alpha$ ($0.1 - 0.4$) balances responsiveness with stability, yielding consistently high AUC across both models. Including "Final answer: " in the prompt slightly AUC across all $\alpha$, which confirms that the prefix tightens the coupling between EAT stabilization and Pass@1 convergence.

### I.4   GPQA-Diamond filtering and error analysis

In the main text, when evaluating EAT's early exiting performance on both versions of GPQA-Diamond ( Fig. 2, last two columns, bottom row), we kept only samples where Pass@1(Avg@128) reaches 0.8 at the end of reasoning. The full results without filtering are shown in Fig. 20, where we do not observe EAT introducing efficiency gain. The reason is that many instances in GPQA-Diamond are either not solvable or the model would show decreasing Pass@1 throughout reasoning. For such questions, the naive token-based early exiting would be preferable to EAT. We present detailed discussion and examples of these two cases below, using questions from open-ended version of GPQA-Diamond.

**Non-solvable question**  Due to the difficulty of GPQA-Diamond, the reasoning model often completely fails to solve some of the questions, regardless of the reasoning efforts. Examples are shown in Fig. 14, where the Pass@1 stays constantly low throughout the reasoning. In such scenarios, EAT does not stabilize; therefore Alg. 1 would use up all $T$ tokens.

**Questions with decreasing Pass@1**   Certain problems begin with a relatively high Pass@1, but the value gradually declines as reasoning continues (Fig. 15). Here EAT often fails to stop at the optimal exit position, which does not show an obvious correlation with EAT.

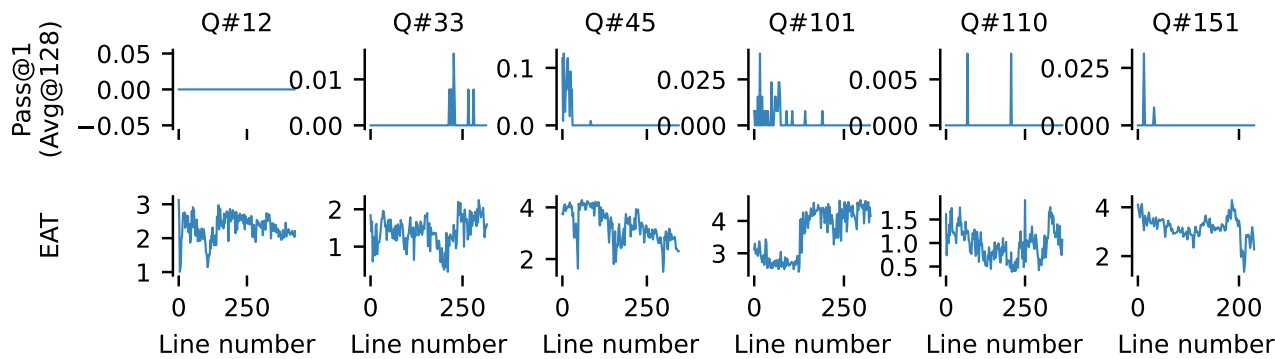

*Figure 14.* On unsolvable question, EAT does not stabilize and therefore would use up all tokens under Alg. 1.

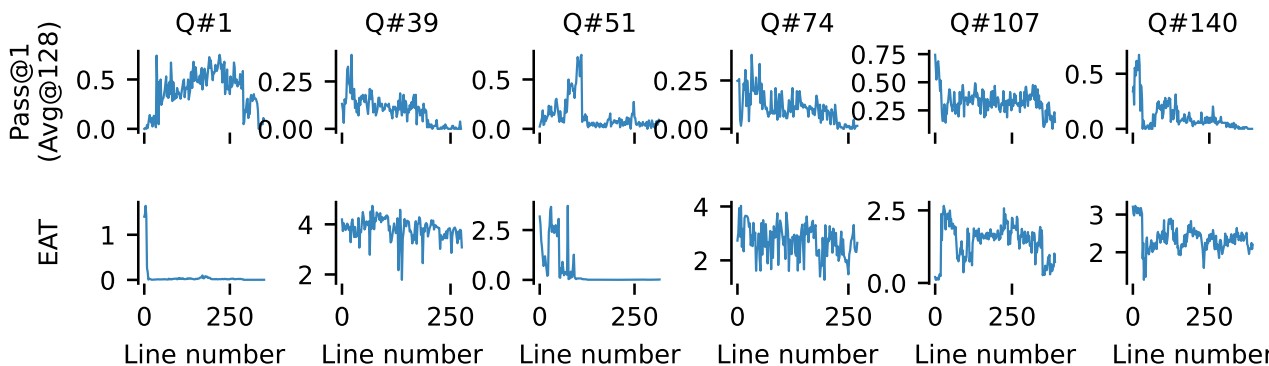

*Figure 15.* On questions with decreasing Pass@1, EAT either does not stabilize or stabilizes at positions where the optimal Pass@1 position has passed.

### I.5  Comparison with one-pass confidence

Here we compare EAT with the confidence score proposed by (Yang et al., 2025b). In particular, given a reasoning trace R, (Yang et al., 2025b) proposes to roll out $T$ tokens (denoted as $\{a_1, \ldots, a_T\}$) *greedily* after R appended with an answer-inducing string ("</think>, Final answer:\n"), and computed the confidence as

$$\texttt{Confidence}(\texttt{R}) = \exp\left( \frac{1}{T} \sum_{t=1}^{T} \log p(a_t \mid \texttt{R}, \{a_{t'}\}_{t' < t}) \right) \tag{16}$$

i.e. the length-normalized likelihood for a single generated rollout. Note that although not discussed in the original paper, the confidence score is equivalent to a single-sample estimation of the $T$-gram entropy over $\{a_1, \ldots, a_T\}$ if we transform it back in the log space and remove the length normalization.

We provide some examples of confidence vs. EAT in Figure 16. Overall, we find that EAT shows similiar correlation with Pass@1 compared with confidence, while confidence is slightly noisier as it is a quantity over multiple tokens.

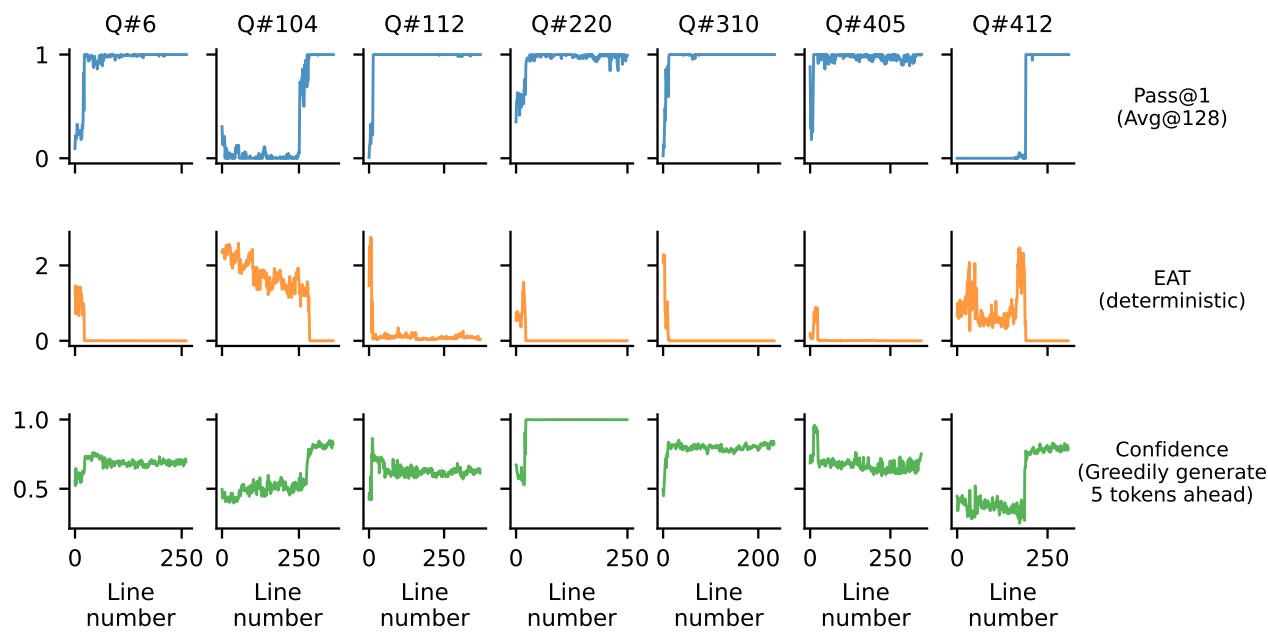

*Figure 16.* Both `EAT` (Eq. (5)) and confidence (Eq. (16)) stabilizes as Pass@1 plateaus.

## I.6 Math500 error analysis

We present examples where the reasoning model fails to solve at the end of reasoning (10,000 tokens) in Fig. 17.

## I.7 Early stopping for Claude 3.7 sonnet

We demonstrate using a locally deployed Qwen3-4B-Thinking-2507 to compute EAT for the reasoning trace received from Claude 3.7 (provided by OpenRouter), an API model, evaluated on the first 8 questions of AIME2025. Note that the API returns the thinking tokens block by block in a streaming way (approximately 5 tokens per block), and we compute `EAT` every time we receive a chunk of 20 blocks.

We show the trace of Pass@1 (Avg@1), `EAT`, and elapsed time in Figure. 18 v.s. chunk index, where the gray dashed line denotes the early stopping point detected by our algorithm, under $\alpha = 0.2$ and $\delta = 10^{-3}$. The results indicate that our algorithms, on solvable problems, save *at least one minute* for each query.

We are unable to further scale up the experiment size due to the significant cost of each run: Each single question requires \$20-\$30 as the experiments need to generate answer rollout at partial reasoning to evaluate early stopping correctness. However we believe the existing results, as well as the large-scale run on Llama-70B, are already sufficient evidence for `EAT`'s compatibility under black-box setting.

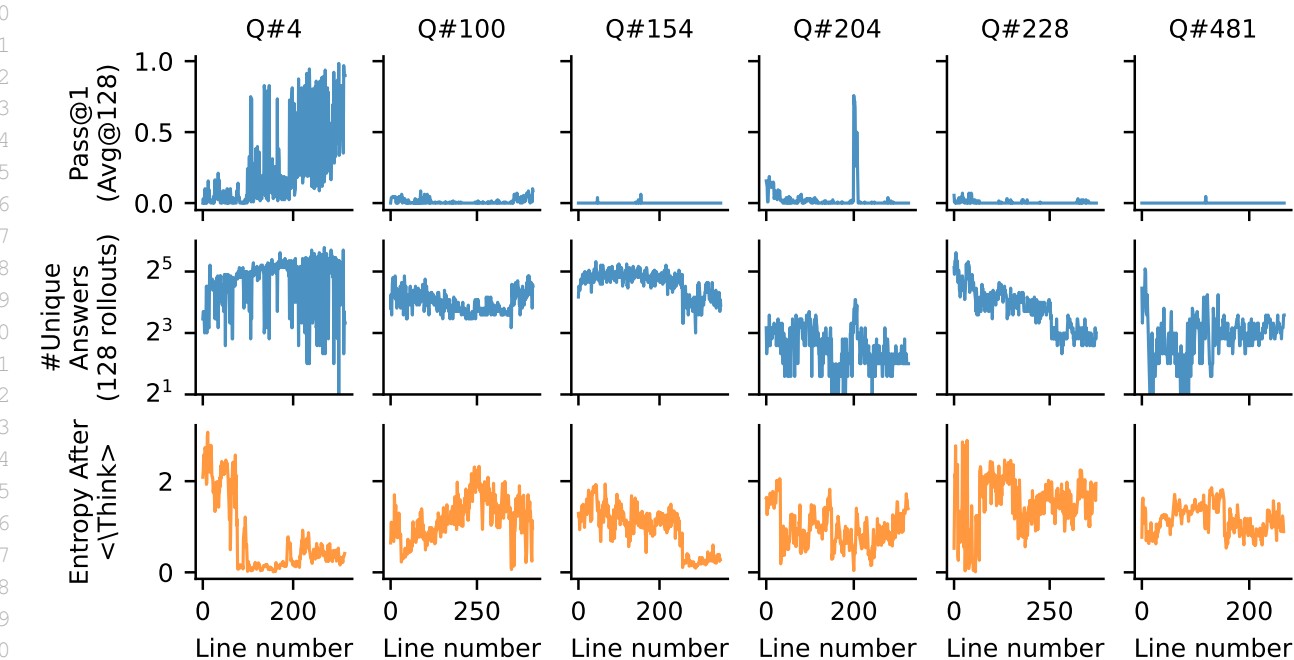

*Figure 17.* On questions from Math500 with low Pass@1 (Avg@128), evaluated on DeepSeek-R1-0528-Qwen3-8B, the number of unique answers as well as `EAT` stays at a high value and perturbs significantly, indicating that the model is struggling to reach a certain and confident answer. Note that the model can potentially solve the problem with more reasoning; therefore, whether these problems should be halted early depends on whether the users want to take the potential risk of premature stopping.

## J    Example of reasoning model outputs

We query one of the newest reasoning models (DeepSeek-R1-0528-Qwen3-8B) with a simple question: "What are the first seven digits of Pi?". The code block below shows the model's output under the recommended decoding strategy, random decoding with a temperature of 0.6 and a top-p value of 0.95. The bold black text shows the question body, and the bold red text denotes the special begin-of/end-of thinking tokens.

The model arrives at the correct answer in its very first paragraph, yet continues to recheck and restate the same conclusion across more than ten subsequent paragraphs without any meaningful revision, which we consider as qualitative evidence of overthinking.

```
What are the first seven digits of Pi?<think>
Okay, the user is asking for the first seven digits of Pi. Let me start by recalling what
Pi is. Pi is the ratio of a circle's circumference to its diameter, approximately
3.1415926535... So the first seven digits would be 3.141592. Wait, I need to count
carefully. Starting from 3, then 1,4,1,5,9,2. That's seven digits after the decimal, but
including the whole number part, it's 3.141592, which is eight digits total. Hmm, maybe
the user means the first seven digits after the decimal? Let me check the question again.
They said "first seven digits of Pi," which could be ambiguous. Commonly, people refer to
the digits after the decimal as the "digits of Pi." But sometimes, the whole number is
included. I should clarify this to avoid confusion.

Wait, the user might be a student or someone just starting to learn about Pi. They could
be working on a project or homework that requires the initial digits. Maybe they need it
for a calculation, programming, or a school assignment. Including the whole number part
would be 3.14159265..., so the first seven digits including the 3 would be up to 3.141592.
 But the question specifically says "digits," not "digits after the decimal." I should
present both possibilities but clarify the standard interpretation.

Also, the user might not know that Pi is an irrational number with infinite digits. They
might just need the first seven for a specific purpose. Are they calculating something
that requires a certain precision? Or maybe they're curious about Pi's properties. Since
```

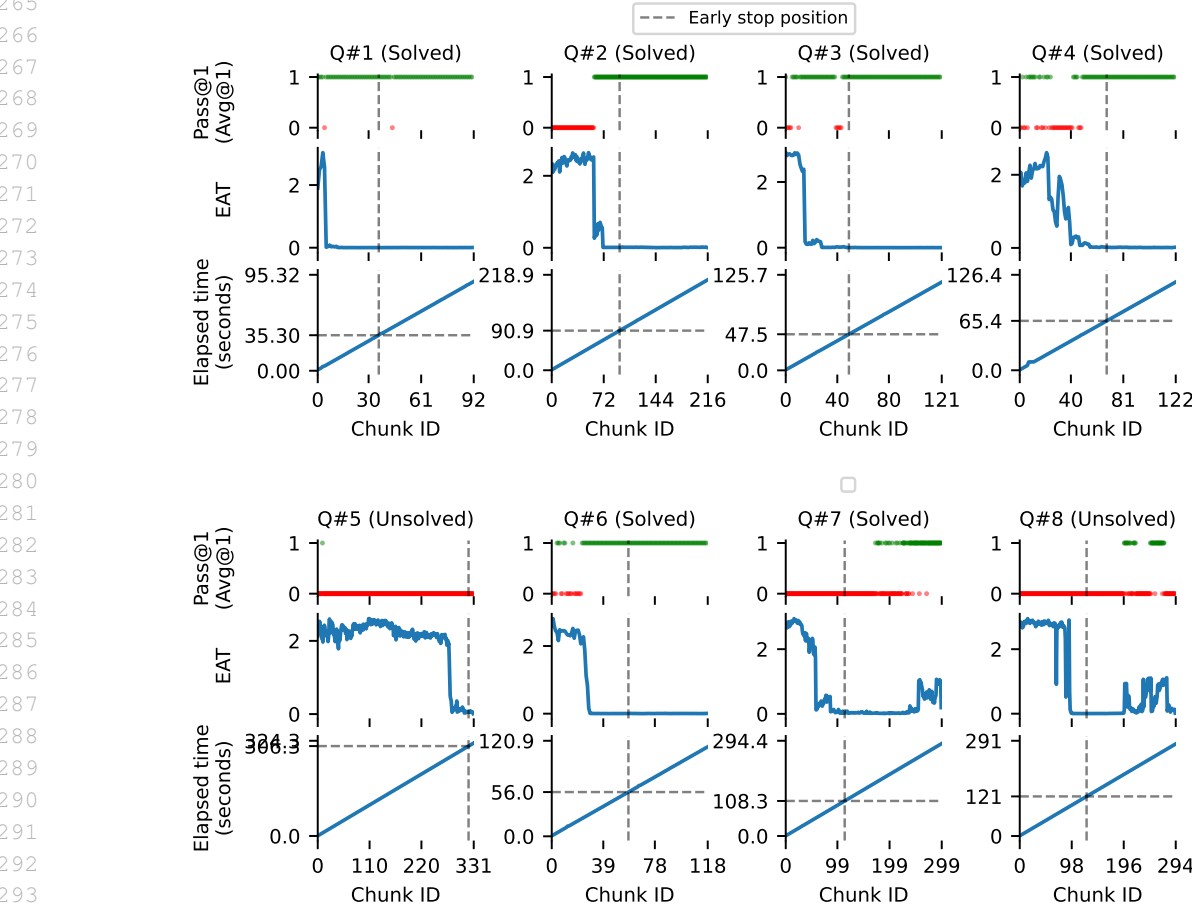

*Figure 18.* On the first 8 questions from AIME2025, we generate reasoning using Claude 3.7 and compute `EAT` with Qwen3-4B-Thinking-2507. The gray dashed line indicates the early stopping position acquired by thresholding the variance at $10^{-3}$. The in brackets (solved / unsolved) denotes the correctness of the final answer included in Claude's auto summarization of the whole reasoning. Our approach successfully early stops all solvable problems except for Q#7, saving at least 60 seconds for each question.

the answer is straightforward, I can provide the digits after the decimal and mention the whole number part if needed. Let me make sure the response is clear and concise, avoiding any technical jargon unless necessary.

Including the 3 would be the first seven digits as a whole: 3,1,4,1,5,9,2. Without the 3, it's seven digits after the decimal: 1415926. But the common way to refer to the first seven digits is usually including the 3. Wait, no. Let me think. If someone says "the first seven digits of Pi," they often mean the initial seven digits, which includes the 3. For example, Pi starts with 3.141592..., so the first seven digits are 3.141592. But the user might be expecting the digits after the decimal.

I should provide both answers but specify which is which. The user's main need is the first seven digits, so I'll present the sequence including the 3 and then the seven digits after the decimal. That way, it covers both interpretations. Also, maybe the user is testing if I know the exact digits or if there's a trick. But since Pi is well-known, the answer should be straightforward.

Let me verify the digits. Pi is approximately 3.1415926535, so the first seven digits are 3.141592. If they want seven digits after the decimal, it's 1415926. But the question doesn't specify, so I'll go with the common understanding. The user might not have specified, but in most contexts, "digits of Pi" includes the integer part.

I'll structure the answer by first giving the seven digits including the 3, then

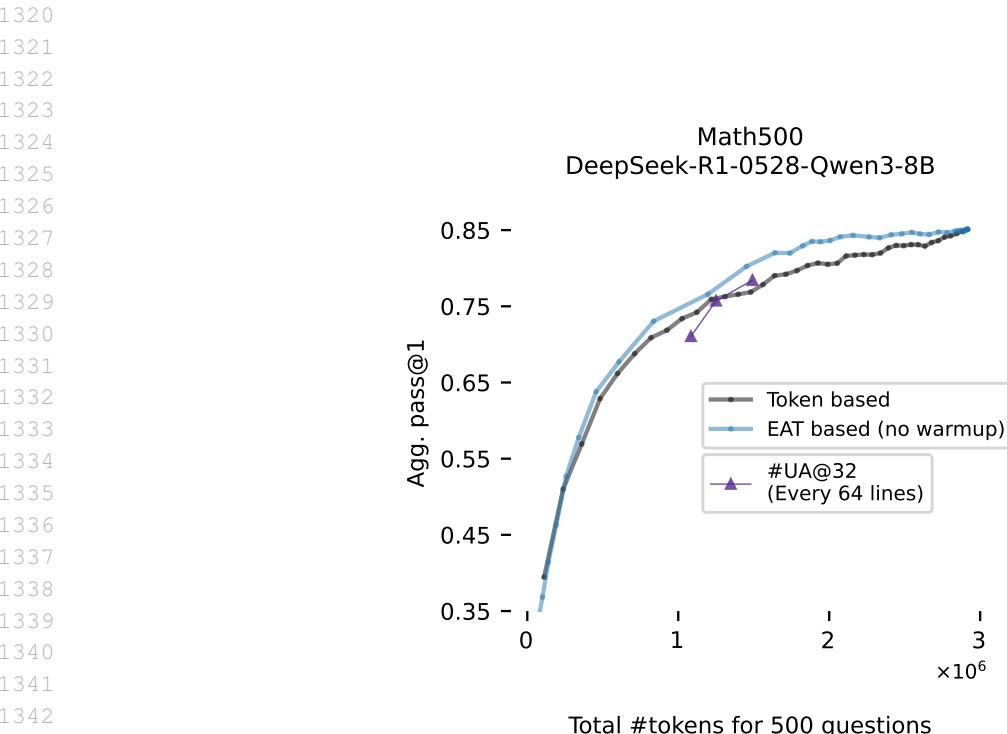

*Figure 19.* Comparision between #UA@32 v.s. EAT under similiar computing budget, where we evaluate #UA@32 every 64 lines to compensate for the overhead of generating 32 rollouts.

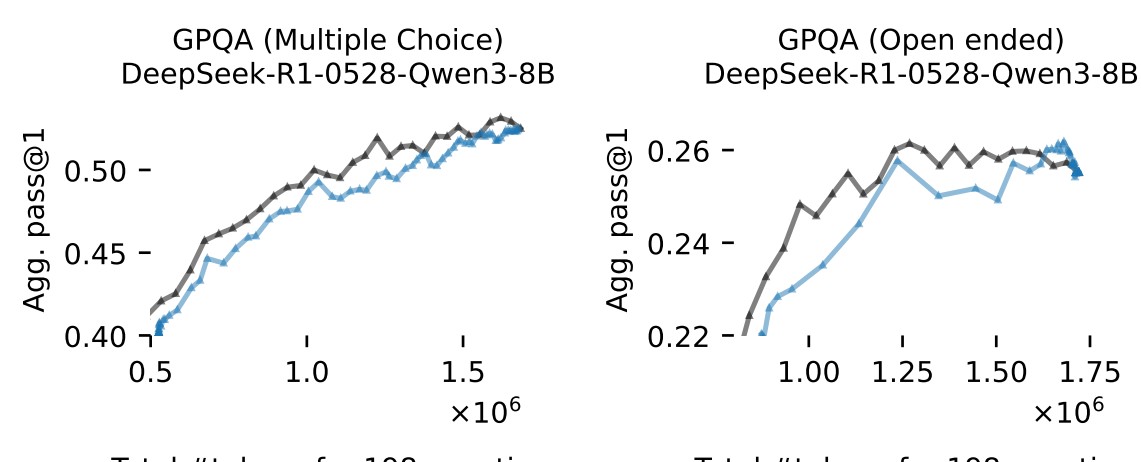

*Figure 20.* Performance of EAT based early stopping on unfiltered GPQA-Diamond.

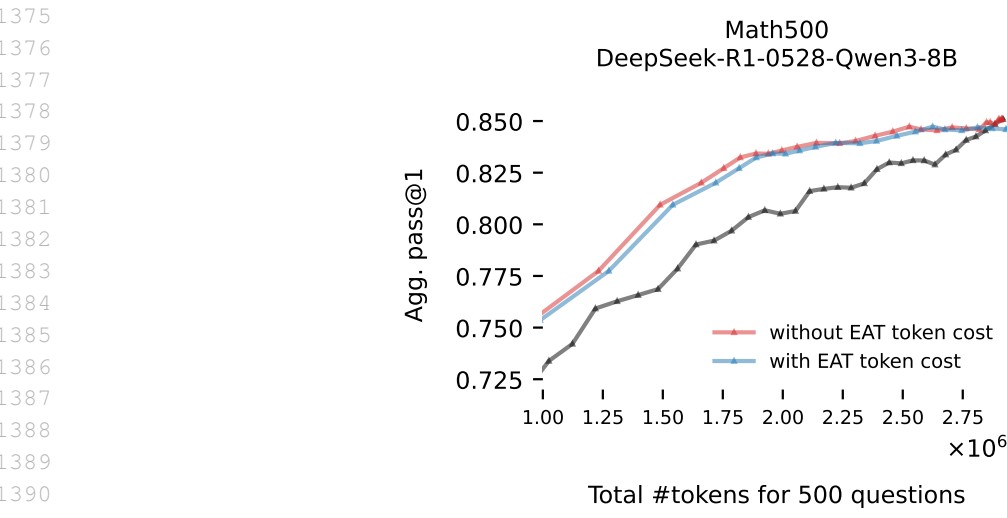

*Figure 21.* Performance v.s. total token usage, where we additionally take into account the overhead introduced by computing `EAT` (counted as 1 token for every evaluation). `EAT` based early stopping still shows efficiency gains, thanks to the minimal overhead of evaluation.

clarifying the seven digits after the decimal. That way, the user gets the information
they need without confusion. Make sure to explain briefly so they understand the
difference.

Also, considering the user's possible deeper needs: they might be preparing for a
presentation, writing a program that needs Pi's digits, or just curious. By providing both
, I address different possible use cases. They might not have realized Pi includes the 3,
so this could help them in the future.

I should mention that Pi is irrational and non-repeating, but that's probably beyond the
scope unless they ask. Just stick to the digits. Keep the answer simple and clear,
avoiding any markdown as per the instructions.

Alright, final answer: first seven digits including the 3 are 3.141592, and the seven
digits after the decimal are 1415926. But the user's question is about the first seven
digits, so I'll present 3.141592 as the answer, then note the seven after the decimal if
needed. Make sure to use the correct terminology so the user knows the difference.

Wait, let me check the sequence again. Pi is 3.1415926535... So the digits are 3, 1, 4, 1,
5, 9, 2. That's seven digits. If they want seven after the decimal, it's 1,4,1,5,9,2,6.
But the initial seven digits are 3.141592. The user might be confused by the decimal point
. Should I exclude the decimal point and just list the digits?

Yes, that's better. The first seven digits are 3141592. The seven digits after the decimal
 are 1415926. But the user asked for the first seven digits, so I'll go with 3141592.
However, the question might be interpreted as the seven digits after the decimal. To be
safe, I can present both but state the main answer clearly.

I think the best approach is to give the first seven digits as 3.141592, then note that
the seven digits after the decimal are 1415926. That covers both bases. The user can
choose based on their need.

So, the answer will be: The first seven digits of Pi are 3.141592. If you're asking for
seven digits after the decimal, those are 1415926.

But let me confirm the count. Including the 3: positions 1-7 are 3,1,4,1,5,9,2. After the
decimal: 1,4,1,5,9,2,6. Yes, that's correct.

I'll make sure the response is clear and addresses both possible interpretations. The user

```
1430    might appreciate the thoroughness.
1431
1432   Also, maybe the user is not aware that Pi is approximately 3.14, but they want more digits
1433   . By providing both, I can help them understand the precision needed.
1434
1435   In conclusion, the answer should be the first seven digits including the 3, which is
       3.141592, and if they need seven after the decimal, it's 1415926.
1436
1437   I'll format the response accordingly, making it easy to read with the digits listed
1438   clearly.
1439
1440   Alright, let me write the response now.
       </think>
1441   The first seven digits of Pi (including the integer part) are **3.141592**.
1442
1443   If you're asking for the first seven digits *after* the decimal point, they are
1444   **1415926**.
```

## K  Prompts

For Math500, AIME2025 and the free-form version of GPQA-Diamond, we used

```
Solve the following math problem efficiently and clearly. Please reason step by step, and
put your final answer within \boxed{}.
```

For the multiple-choice version of GPQA-Diamond, we used

```
Solve the following multiple choice problem efficiently and clearly. Please reason step by
 step, and return one single option as the answer, put your answer within \boxed{}.
```

When evaluating the open-ended version of GPQA-Diamond, we check the correctness using LLM as a judge following the prompt provided by Chandak et al. (2025)

```
Your task is to judge whether the given response to a question matches a given ground
truth answer or not. You are provided with a question, a ground truth response, and the
response you need to judge.
For a response to "match", it must have at least as much information as the ground-truth.
The response can have more information than the ground-truth. It can be more specific (for
 example, "Labrador" is more specific than "dog"), or have additional possible correct
answers. But it must cover everything mentioned in the ground-truth. It is okay if it
covers it in different words, i.e. paraphrased.
For numeric answers, the relative error, defined as |response - ground truth| / mean(
response, ground truth), must be less than 1% for the response to be judged as a correct
match. Here, if the ground truth is a specific numeric quantity but the response is a
range, then they don't match (even if the range contains the ground truth).

Possible judgments:

"0": The response does not match the ground-truth answer.
"1": The response matches the ground-truth.

Question: "{question}"
Ground truth: "{target}"
Response: "{response}"

Your job is to ONLY check whether the given response matches the ground truth answer or
not in the context of the question. You DO NOT NEED to assess the correctness of the
response. This is part of an automated evaluation process, therefore you MUST OUTPUT your
final answer as "0" or "1" in <answer> </answer> tags.

Think step by step and end your response with <answer>0</answer> OR <answer>1</answer>
TAGS.
```

