# OpenReview forum: "EAT: Entropy After $\langle \texttt{/Think} \rangle$ for reasoning model early exiting"
_ICML.cc/2026/Conference — Submitted to ICML 2026_

### Official Review · Reviewer_fkVq · 2026-03-02

**Soundness:** 3
**Presentation:** 2
**Significance:** 3
**Originality:** 2
**Overall Recommendation:** 4
**Confidence:** 3

**Summary:**

This paper studies overthinking in reasoning language models and shows that answer distributions often converge well before the reasoning process finishes, leading to wasted computation. To address this, the authors propose Entropy After  (EAT), a lightweight signal that measures the entropy of the first answer token when a temporary </think> token is appended during reasoning. They introduce an early-exit rule based on the variance of this entropy, enabling adaptive stopping without additional training or expensive rollouts. Experiments across multiple reasoning models and benchmarks demonstrate that EAT reduces reasoning token usage (up to ~20%) while maintaining accuracy, and can also be applied in black-box settings using a proxy model.

**Compliance With Llm Reviewing Policy:**

Affirmed.

**Final Justification:**

The authors have adequately addressed my concerns, providing additional experiments on MMLU, insightful discussion on multi-token entropy, and clarifications on black-box applicability and comparisons with recent baselines.

I maintain my original score of Weak Accept, as the paper presents a simple yet practical early-exit signal with broad applicability, despite limitations in evaluation scope and theoretical justification.

**Key Questions For Authors:**

1. Evaluation on Simpler Benchmarks (e.g., MMLU). Most experiments are conducted on math-heavy or structured reasoning benchmarks. Have you considered evaluating EAT on datasets such as MMLU? Since many MMLU questions are relatively short and may not require long reasoning chains, this setting might more clearly demonstrate the effectiveness of adaptive early stopping on simpler tasks.
2. Applicability to Hybrid-Thinking Models. The experiments focus on reasoning models with explicit <think> ... </think> structures. Have you tested EAT on hybrid-thinking models (e.g., models that dynamically switch between no-think and think modes)? It would be interesting to understand whether EAT remains effective when reasoning behavior is not strictly structured.
3. Single-Token vs. Multi-Token Entropy. EAT relies on the entropy of the first answer token after </think>. Have you explored using multi-token entropy (e.g., over the first few answer tokens) as a potentially more robust signal? This might improve stability, particularly for tasks with longer or more variable answer formats.
4. Comparison with Recent Early-Exit Methods. There have been several recent works on early stopping in reasoning models (e.g., Dynamic Early Exit in Reasoning Models). Could you include comparisons with more recent early-exit approaches to better contextualize the advantages of EAT?

**Limitations:**

yes

**Strengths And Weaknesses:**

- Strengths
	- Simple and Training-Free Method. The proposed EAT signal is conceptually simple, requires no additional training, and introduces only two lightweight hyperparameters, making it easy to implement and integrate into existing reasoning systems.
	- Computational Efficiency. Unlike rollout-based uncertainty or confidence methods, EAT avoids expensive answer sampling and adds minimal overhead, enabling meaningful reductions in reasoning tokens (up to ~20%) without sacrificing accuracy.
	- Broad Applicability (Including Black-Box Settings). The method can operate both in white-box settings (using model logits directly) and in black-box scenarios via a smaller proxy model, increasing its practical relevance for large API-based reasoning models.

- Weaknesses
  - Potential Weakness on Long-Form Outputs. The method relies on the entropy of the first answer token, which may be less informative for tasks requiring long-form outputs (e.g., code generation or multi-paragraph explanations), where correctness depends on many downstream tokens.
  - Evaluation Scope Concentrated on Structured Reasoning Benchmarks. Most experiments are conducted on math and structured QA benchmarks, where answers are relatively short and well-formatted. It remains unclear how the method performs in more diverse or real-world reasoning scenarios, particularly those with higher output variability or less deterministic answer formats.
  - Limited Evaluation on Explicit Reasoning Models Only. The experiments are conducted only on reasoning models that use explicit < think> ... < /think> tokens. It is unclear whether EAT would work for hybrid-thinking models. Therefore, the generality of the method beyond structured reasoning models is not fully demonstrated.

---

> ### Author Rebuttal · Authors · 2026-03-30
>
> # MMLU
> Thanks for raising this good experiment setting!
>
> We test EAT on 60 questions from MMLU from `Astronomy` and `global_facts`.
> The Pass@1 (Avg@1) v.s. EAT figures for selected questions are shown at https://imgur.com/a/ojrwPmM and https://imgur.com/a/NcGED1q where we observe correlation between Pass@1 stability and EAT decrement + stability.
>
> To use EAT for early stopping on MMLU, we adopt a strategy even simpler than thresholding EMA estimated variance: Since MMLU is a multiple-choice dataset, we can use a handcrafted condition: $2^{\text{EAT}_t} < 2 - \epsilon$.
> The idea is grounded in interpreting the exponential of entropy (computed on base 2 in our cases) as the *number of effective choices*, so when it goes below 2, we know the model is fairly confident in one option.
>
> Based on this observation, we stop the reasoning when the value of $2^{\text{EAT}_t}$ averaged over a sliding window of 3 steps goes below a threshold near $2$. The results are presented below:
>
> | $2^{\text{EAT}}$ threshold | No threshold | 2       | 1.9     | 1.8     |
> |-----------------|------------------|---------|---------|---------|
> | #tokens         | 63449            | 41318   | 43325   | 44022   |
> | correcntess        | 40 / 60          | 38 / 60 | 39 / 60 | 40 / 60 |
>
> where we again see significant token reduction with little or no accuracy hurt.
>
> Due to the limited time frame of the rebuttal, we can only test 60 questions; we are testing more questions on MMLU for the revision.
>
> # Hybrid thinking models
> To the best of our knowledge, hybrid thinking models are those that you can turn on and off thinking mode with a hard switch (e.g. Qwen3 https://huggingface.co/Qwen/Qwen3-30B-A3B#switching-between-thinking-and-non-thinking-mode).
>
> It is unclear to us which existing open-source models support dynamic interleaving between thinking and non-thinking modes when answering a query.
> We are happy to study EAT's applicability on such models if the reviewer can point to us to any model with such a feature.
>
> # Multi-token / gram entropy
> This is a very good question! It is an interesting but complicated one.
>
> ## Compute of multi-token / gram entropy
> Firstly, computing multi-token entropy is hard. It can only be computed with Monte Carlo estimation by generating rollout after `</think>`, as is discussed in Lines 678 and 686 in section C, Appendix in our current manuscript.
>
> This implies that:
> - You need to trade off between the variance of the estimation and overhead of computing the metric.
> - The decoding hyperparameter will affect the value and variance of the estimation.
> - A significant implementation complexity and overhead will be introduced, as parallel batch decoding is non-trivial to implement while generating rollout is slow.
>
> ## Performance of multi-token entropy
> Regardless, we do have some results for multi-gram entropy on Math500 (https://imgur.com/a/BHplA5Z).
> We estimate 5-gram entropy with 10 rollouts, measure the performance by the area under accuracy-token curve under various EMA $\alpha$.
> Broadly, we find 3-gram entropy can work better than unigram entropy, but the performance starts to drop with more grams, which we believe is caused by the finite sample error.
>
> We believe that if the computational trade-off can be handled properly, multi-token entropy can serve as a more informative and powerful signal compared with the unigram EAT if estimated accurately.
>
> ## Long-term generation
> However, in real long-term generation, e.g., coding, even the first ten tokens may all correspond to the import statements, irrelevant to the actual solution, so it remains unclear how many future steps we need to look at.
> Therefore, we believe some tweak of the proxy model approach can be more suitable for such a purpose (as we discussed in the second paragraph in the limitation section), which can provide an early stopping signal with more controllable overhead.
>
> Perhaps a broader challenge here is the generalizability of  "stop when certain/confident" principle: For free-form generation (e.g. creative writing), there may not exist a correct answer, so the model can be uncertain and correct; Or consider the coding setting, even the model is uncertain and generates a diverse set of solutions, we can still pick the correct answers with unit tests (therefore people often use Pass@K for code generation)
> Although not discussed in the reasoning model domain, there have been works arguing the pitfall of such a principle in long-horizon learning-to-defer settings (https://arxiv.org/abs/2307.02764).
>
> Regardless, we have included an extended discussion on multi-token entropy in our revised draft!
>
> # Comparison with other methods
>
> We actually already included a comparison with Dynamic Early Exit in Reasoning Models in **Figure 4 in our original submission** on Math500!
>
> The main observation is that EAT shows performance comparable to the confidence score they proposed, while being **5 times cheaper** for uncertainty signal evaluation.

---

> > ### Author Rebuttal · Reviewer_fkVq · 2026-04-02
> >
> > the authors sufficiently adressed the issues. I maintain my score.

---

> > > ### Author Response · Authors · 2026-04-03
> > >
> > > We would like to thank the reviewer again for the positive, detailed, and constructive feedback.
> > >
> > > We have incorporated all additional results and contents into our manuscript, and we believe it strengthens our work significantly!
> > >
> > > We are happy to address any further concerns from the reviewer during the rebuttal time window.
> > >
> > > Thanks,
> > >
> > > Authors

---

### Official Review · Reviewer_45Au · 2026-03-08

**Soundness:** 3
**Presentation:** 2
**Significance:** 3
**Originality:** 1
**Overall Recommendation:** 2
**Confidence:** 5

**Summary:**

This paper studies inefficiencies in reasoning LLMs caused by excessive chain-of-thought generation (“overthinking”). The authors empirically observe that the correctness of the predicted answer often stabilizes early in the reasoning trajectory. Motivated by this observation, they propose Entropy After ```</think>``` (EAT), a lightweight signal for early exiting reasoning. The key idea is to append a hypothetical </think> token during reasoning and compute the entropy of the next-token distribution. When the entropy stabilizes (measured via an EMA variance threshold), the reasoning process is terminated and the final answer is generated.

**Compliance With Llm Reviewing Policy:**

Affirmed.

**Final Justification:**

While I appreciate the clarifications, the rebuttal mainly reframes premature stopping as a threshold-selection issue without providing additional experiments on reliability or calibration, and the core assumption about entropy remains largely empirically motivated without strong justification; therefore, I maintain my original score.

**Key Questions For Authors:**

1. How sensitive is the method to decoding randomness? Since the signal is computed along a single reasoning trajectory, could different sampled reasoning paths lead to different stopping behavior?
2. The current evaluation mainly focuses on aggregate token–accuracy trade-offs. However, this does not directly reveal whether the early stopping decisions themselves are reliable. It would be helpful to analyze the distribution of early stop positions and quantify the rate of premature stopping. Such analysis would provide a clearer understanding of whether the stopping criterion is well calibrated.

**Limitations:**

Yes

**Strengths And Weaknesses:**

## Strengths
1. **Addresses an important practical problem.** Reducing inference cost for reasoning LLMs is an important and timely problem, especially with the emergence of test-time scaling techniques.
2. **Simple and easy to implement.** The proposed signal is extremely lightweight: it requires only one additional forward pass and does not require sampling rollouts, additional models, or training.

## Weaknesses
1. **Limited novelty.** The proposed method essentially consists of measuring the entropy of the next-token distribution after appending a </think> token and using a threshold on its stability as a stopping rule. Conceptually, this is a very simple heuristic uncertainty signal.

2. A central assumption of the method is
> the entropy of the next-token distribution after </think> correlates with reasoning convergence.

However, the paper provides little theoretical justification for why this specific signal should reliably indicate that further reasoning is unnecessary. In particular, the entropy is computed over the first token after </think>, which is often a stylistic token (e.g., “Final”, “The”, “Thus”) rather than the answer token itself. It is therefore unclear whether this entropy truly reflects answer uncertainty. A stronger theoretical or probabilistic explanation of this phenomenon would significantly strengthen the paper.

3. **Limited comparison with alternative signals.** The paper mainly compares EAT with token-budget stopping and rollout-based uncertainty measures. However, other natural baselines could be considered, such as LL [1] and DeepConf [2]. A more systematic comparison with alternative uncertainty signals would strengthen the empirical evaluation.


[1] Murray K, et al. Correcting length bias in neural machine translation

[2] Fu Y, et al. Deep think with confidenceIs

---

> ### Author Rebuttal · Authors · 2026-03-30
>
> # Novelty and simplicity
> We agree there are many concurrent works along this line. However, no other works have demonstrated rollout-free and black box model (e.g., Claude 3.7) compatibility, a strength acknowledged by reviewers qBAY, 2EwS and fkVq.
>
> We also believe simplicity is a major advantage of EAT and conducted a broad set of ablation studies, e.g. Appendix G,I (acknowledged by reviewer qBAY), to confirm its effectiveness. We also make it clear the limitations (Sec.6) on when a simple method can fail and more sophisticated approaches may be needed. As we address next, EAT is theoretically motivated too.
>
> # Theoretical explanation
> Our approach is based on maximizing the information gain of the conclusions of reasoning with respect to the length of the reasoning chain. This principled strategy is widely used for data selection to reduce model uncertainty. In reasoning, the "data" we optimize are the additive perspectives gained through further reasoning (see Figure 7 for an EAT vs. reasoning-steps comparison). A direct application of information gain would pose high computational costs. Therefore, an important finding we share in the paper is that the entropy over the first token is an informative proxy for model uncertainty.
>
> Namely, we looked at the trace of 5-gram entropy (estimated with 10 rollouts) v.s. unigram entropy. The result suggests that when the model is certain about the first token, randomness in the future 5 tokens also stabilizes: https://imgur.com/a/VXoxEmz
> We also see this through some rollout examples at positions with high and low EAT value:https://imgur.com/a/QeQ7P3J
> Taken together, this evidence highlights that the first token is as a critical anchor token that decides the content of the full rollout, a behavior that potentially arises from the post-training
>
> However, we clarify that for older models, such as DeepSeek-R1-Qwen1.5B, the entropy directly after `</think>` is not very informative: We need to append a prefix and look at next token entropy after `</think> Final answer: `.
> Appendix D and I.3 provided a detailed ablation and discussion on the prefix.
> (Note that the prefix is identical to the one we used for eliciting an answer from a partial reasoning chain).
> Empirically, we also noticed that containing this prefix leads to better early stopping performance.
> Additionally, in the tool calling setting (Appendix I.2), we include a prefix of "[", as the model would always output "[" after `</think>` per the instructions, so the entropy over the first token after `</think>` is not informative.
>
> Lastly, in the current manuscript (App. C), we acknowledge the gap between the empirical observations and the theory, but we believe these novel observations can motivate future theoretical work on reasoning mechanisms.
>
> # Comparison with alternative signals
> In the current manuscript, we have compared against confidence score [1] in Fig. 4.
>
> LL from Murray K, et al., if we understood correctly, corresponds to length-normalized likelihood, which is identical to the confidence signal [1] (see Page 21 in our manuscript for explanation)
>
> For DeepConf, it is unclear to us how to perform early stopping with their proposed confidence score accumulated inside a reasoning chain, which they used for weighting parallel reasoning chains rather than for early stopping.
>
> [1] Dynamic Early Exit in Reasoning Models
>
> # Questions
> ## Decoding randomness
> In principle, the optimal length of reasoning trajectory depends on decoding randomness because answer accuracy depends on the sampled reasoning trajectory. This can be seen with these examples under 4 different seeds for questions on Math500 -- the Pass@1 accuracy saturates at similar points, but with a few obvious exceptions:
> https://imgur.com/a/rCbLdYL
>
> In this sense, it is important for any early exit method to be sensitive to decoding randomness. In the above plots, EAT is indeed responsive to the variation in optimal reasoning length, stabilizing earlier (or later) when the underlying Pass@1 accuracy stabilizes earlier (later, respectively).
>
> ## Premature stopping
> Note that the stopping criterion and the degree of premature stopping depend on the threshold value $\delta$. Dots on Fig 3's lines denote different threshold values:
> - Dots on the right have a stricter threshold; we do not have premature stopping (no accuracy loss compared with full budget).
> - Dots on the left have a looser threshold; we use fewer tokens, but we also make more mistakes.
>
> Note that calibration of the stopping criterion is not the focus of our work: We aim to show that early stopping with EAT is consistently more token efficient than baseline approaches through enumerating over thresholds.
> To calibrate the criterion, we need a separate validation set and a user-specified target risk to calibrate against. With these two ingredients, we can utilize distribution-free risk control framework [1] to find a calibrated stopping criterion.
>
> [1] https://arxiv.org/abs/2602.03814

---

> > ### Author Rebuttal · Reviewer_45Au · 2026-04-03
> >
> > Thank you for the rebuttal. While I appreciate the clarifications, my core concerns remain unaddressed. In particular, the central assumption that *the entropy of the first token after `</think>` reflects reasoning convergence* remains largely an empirical observation without clear justification. The response also reframes premature stopping as a threshold trade-off, but does not evaluate the reliability or calibration of the stopping decisions themselves.
> >
> > Overall, I do not find that the rebuttal resolves the key issues regarding novelty, justification, and evaluation. I will therefore maintain my original score.

---

> > > ### Author Response · Authors · 2026-04-06
> > >
> > > We would like to thank the reviewer for the acknowledgement of our rebuttal.
> > >
> > > --------------
> > >
> > > > The response also reframes premature stopping as a threshold trade-off, but does not evaluate the reliability or calibration of the stopping decisions themselves.
> > >
> > > We would like to clarify that reliability, calibration, and rate of premature stopping are three problems centered around the generic **"how to pick a threshold"** problem for early stopping, a problem orthogonal to the design of stopping signals (see [1, 2, 3] for more extensive discussions, and our extended discussion section below).
> > >
> > > Our submission focuses on developing a new early stopping signal, and we showed that the signal is universally more efficient than the baseline **across all thresholds** (Figure 3).
> > >
> > > We recognize that terminology in this emerging field can vary. If the reviewer is referring to a different formalization of calibration or reliability, we would welcome the opportunity to provide further analysis and build a better understanding of EAT.
> > >
> > > [1] Wang, Xi, et al. "Conformal Thinking: Risk Control for Reasoning on a Compute Budget." arXiv preprint arXiv:2602.03814 (2026).
> > >
> > > [2] Zhou, Cai, et al. "Online Reasoning Calibration: Test-Time Training Enables Generalizable Conformal LLM Reasoning." arXiv preprint arXiv:2604.01170 (2026).
> > >
> > > [3] Wu, Menghua, et al. "Thought calibration: Efficient and confident test-time scaling." Proceedings of the 2025 Conference on Empirical Methods in Natural Language Processing. 2025.
> > >
> > > --------------
> > >
> > > > the central assumption that the entropy of the first token after </think> reflects reasoning convergence remains largely an empirical observation without clear justification
> > >
> > > We have provided justifications from multiple perspectives:
> > > (1) Reasoning model's uncertainty often converges early throughout reasoning, EAT, as a cheap and black-box-model compatible uncertainty signal motivated by information gain, can help determine early stopping. (2) In the rebuttal, we provide evidence showing that the entropy over the first token is informative and predictive of the uncertainty in the full answer string.
> > >
> > > We have also openly admitted the limitations of our justifications (Sec.C in the appendix): To actually have a rigorous proof on why EAT works, one would need to answer the three fundamental open questions about the reasoning model
> > > - Why does the anchor token (e.g. the first token) exist in the reasoning model outputs?
> > > - Why does the reasoning model become more confident throughout thinking?
> > > - Why is the reasoning model's uncertainty implicitly calibrated, e.g. higher uncertainty typically indicates less chance of correctness (Fig.14, 15 in appendix) and vice versa.
> > >
> > > --------------
> > > ## Extended discussion on calibration and reliability
> > >
> > > Recent works (e.g. [1] [2]) that study calibration and reliability in the context of reasoning model early stopping consider a setting where the user would provide
> > > - A labeled validation set composed of per-reasoning-step correctness annotations.
> > > - A target tolerance level ($\epsilon$) for incorrect premature stopping.
> > >
> > > Then an algorithm picks a **threshold** to construct an early stopping criterion that meets $\epsilon$. The quality of such an algorithm is often evaluated from the following two perspectives:
> > > - Calibration: Does the threshold value returned by the algorithm actually reach the target $\epsilon$?
> > > - Reliability: Is the calibration guarantee transferable across datasets and robust to the validation set's finite sample noise and distribution shift at test time?
> > >
> > > We agree that how to pick the early stopping threshold is important for practical deployment, a problem we discussed with reviewer qBAY during rebuttal, but this is a problem orthogonal to the design of stopping signals: **Even under non-adaptive token-based early stopping, the problems still remain.**

---

### Official Review · Reviewer_2EwS · 2026-03-13

**Soundness:** 3
**Presentation:** 3
**Significance:** 2
**Originality:** 2
**Overall Recommendation:** 3
**Confidence:** 3

**Summary:**

This paper proposes a training-free early stopping method to stop overthinking in reasoning models. The idea is to monitor the entropy after </think> (EAT). The authors show that when EAT stabilizes, reasoning accuracy typically saturates, so that the reasoning model can terminate reasoning early. Experiments show that the method can reduce token usage by around 12–22% without degrading accuracy.

**Compliance With Llm Reviewing Policy:**

Affirmed.

**Key Questions For Authors:**

1. The paper reports reductions in token usage. Can you provide more analysis on how latency or inference cost is reduced in real deployment settings?
2. How sensitive is the performance to the choice of proxy model, when choosing to use a proxy model to estimate entropy?

**Limitations:**

Yes

**Strengths And Weaknesses:**

Strengths:
1. EAT is a simple and training-free method to stop overthinking for reasoning models. The idea of monitoring entropy after </think> is also intuitive.
2. The authors show that the method could also be applied in a black-box setting by using a small model as a proxy to monitor entropy. This trick makes the proposed method more practivcle.

Weaknesses:
1. The idea of using entropy as a signal for early stopping reasoning is conceptually similar to many prior approaches in this domain. EAT mainly changes where the entropy is measured (after </think>), making the novelty of this paper limited.
2. The method relies on the empirical observation that entropy after </think> stabilizes when reasoning accuracy saturates. However, the paper does not provide a theoretical explanation for why this signal should consistently correlate with reasoning completion across models and tasks.
3. The reported efficiency gains are relatively modest compared to some related methods. In addition, the empirical comparison is somewhat narrow. The experiments mainly compares EAT against simple baselines such as fixed token budgets or rollout-based heuristics, but fail to evaluate against several recent early-stopping methods.

---

> ### Author Rebuttal · Authors · 2026-03-30
>
> # Weaknesses
>
> ## Novelty
> Yes, we agree that using entropy, or more generally, uncertainty for early stopping and adaptive computing is a commonly adopted idea, as we discussed at the start of Sec 2, where we noted that a similar principle has been adopted since the LSTM days.
>
> The point of departure for our work is to target information gain (Eq. 6) in a no-rollout setting with minimal requirements around access to logits or hidden states in large reasoning models. Typically, information gain is used to acquire informative *data* points, we offer a new perspective on treating reasoning insights as information w.r.t. the final answer and offer a way to optimize it with a simple implementation (a property also acknowledged by reviewer fkVq). This is a significant insight that we believe naturally leads to further research in information-guided reasoning.
>
>
>
> ## Theoretical explanation
>
> Information gain measures the reduction in the number of answer configurations given a partial reasoning trajectory. In principle, as the reasoning trajectory gets longer and more redundant, the gain saturates.
>
> Empirically, we see exactly this.
> Our paper presents extensive results confirming that Pass@1 accuracy and number of unique answers over 128 rollouts (both serve as an empirical measurement of uncertainty, i.e. information gain) stabilize early during the reasoning, providing one of the first empirical confirmations that early stopping leveraging information gain is a useful and robust approach (See Appendix B for comparisons with prior works' emprical evidences).
> See Appendix C for a detailed discussion on this observation-theory connection, as well as the release of a large-scale rollout dataset that we intend to support future theory research for understanding reasoning models.
>
>
> ## Baseline
>
> We did conduct comparisions with DEER [1] in Figure 4 in the original submission, where we observed that EAT shows performance comparable with DEER while being 5 times cheaper for evaluating the uncertainty signal.
>
> It is also worth noting that the efficiency gain number (e.g. 12–22%) we reported is the worst-case scenario where **no accuracy** is sacrificed: If we look at the blue lines with lower accuracy in Figure 3, the efficiency gain is much more significant.
>
> [1] Dynamic Early Exit in Reasoning Models.
>
> # Questions
>
> ## Latency
> In the original submission, we discussed this important question from two perspectives:
> - In black box mode (Figure 5), EAT's compute can overlap with model rollout. When tested on Claude 3.7, EAT saves at least 1 minute for the instances we looked at.
> - In non-black-box settings, Figure 21 (page 26) in the appendix takes into account the actual overhead (measured by token) from EAT evaluation, where every EAT eval is counted as one extra token (sec 4.3).
> We can see that the actual inference cost for reaching target accuracy is still much lower than token-based baseline.
>
>
> ## Sensitivity to proxy model
> We did consider proxy model of various sizes (1.5B, 4B, 8B) in our current experiments, e.g. Fig.3 bottom left, Fig.11 blue v.s. red line. Surprisingly, we did not observe larger models outperforming smaller ones in a statistically significant way. We hypothesize that the reason is that monitoring the reasoning chain is a task much easier than generating one, so small models can be as good as larger ones.

---

> > ### Author Rebuttal · Reviewer_2EwS · 2026-04-04
> >
> > The rebuttal provides helpful clarifications, particularly on efficiency and proxy models. However, my main concerns remain largely unchanged. The novelty still appears limited compared to prior entropy/uncertainty-based approaches, and the justification remains largely empirical. Overall, my assessment remains unchanged.

---

> > > ### Author Response · Authors · 2026-04-07
> > >
> > > We would like to thank the reviewer for the acknowledgement and for acknowledging our approach as intuitive, and our novel black box compatibility as making "the proposed method more practical."
> > >
> > > We are also glad to see that the concerns on *"...the reported efficiency gains are relatively modest compared to some related methods.." and "...the empirical comparison is somewhat narrow..."* are resolved.
> > >
> > > > The novelty still appears limited compared to prior entropy/uncertainty-based approaches,
> > >
> > > We would appreciate it if the reviewer could provide more references for the "prior approaches". We agree that there may exist **concurrent works or works after us** that consider similiar approach (as you said, this is an intuitive approach), therefore, more detailed references would help us clarify the relationship between us and other approaches.
> > >
> > > Lastly, we would also like to highlight that our contribution is *beyond just EAT itself*, our work also provides the following novel contribution
> > > - Massive rollout dataset for future research on reasoning models.
> > > - Extensive empirical results on the reasoning model's uncertainty convergence, verifying the legitimacy of uncertainty-based early stopping.
> > > - Evidence on the monitorability and transferability of reasoning traces across models.
> > > - Evidence of the failure pattern of the "stop when confident" principle on unsolved problems, which has already inspired future works on complementary thresholding mechanisms.
> > >
> > > We believe these results can inspire and motivate future research on better early stopping approaches and understanding the fundamental mechanisms of the reasoning model.

---

### Official Review · Reviewer_qBAY · 2026-03-13

**Soundness:** 3
**Presentation:** 3
**Significance:** 4
**Originality:** 4
**Overall Recommendation:** 6
**Confidence:** 4

**Summary:**

EAT (Entropy After </Think>) proposes a lightweight early-exit signal for reasoning models. The core observation is that if you forcibly insert a </think> token at regular intervals during reasoning and measure the entropy over the single token that follows, the resulting entropy trajectory decreases monotonically and stabilizes. This stabilization point closely coincides with where Pass@1 saturates. Based on this, the authors propose tracking the variance of the EAT trajectory under an exponential moving average, and triggering early exit when the variance drops below a threshold $\delta$. The computational overhead is minimal, just with one forward pass, and the method works in black-box settings, a small proxy model (1.5B) can monitor a much larger reasoning model (70B) without access to its logits. On MATH-500 and AIME-2025, EAT reduces token usage by 12–22% without hurting accuracy.

**Compliance With Llm Reviewing Policy:**

Affirmed.

**Final Justification:**

Thank you for the author's detailed response, which has fully addressed my concerns. Although this work originates from a seemingly intuitive and modest insight, it delivers a thorough and rigorous analysis—encompassing a well-articulated motivation, meticulously designed methodology, comprehensive experiments, and a candid discussion of limitations. The paper is both intellectually rewarding and highly inspiring to read. Accordingly, I have decided to revise my score upward.

**Key Questions For Authors:**

1. EAT brings no efficiency gains on the full dataset, especially for can't sovled question. In practice, a user would not know in advance whether a problem is solvable by the model. Could the authors clarify how EAT should be used in a realistic deployment setting where no such pre-filtering is possible? Is there a way to detect at inference time whether EAT is likely to be a reliable early-exit signal for a given problem? One potential direction would be to combine EAT with a complementary "stagnation" signal — for instance, if EAT's variance fails to decrease within a certain number of reasoning steps, the model could be forced to exit early regardless. This would handle the unsolvable case without requiring any pre-filtering. That said, this is just a rough idea that might be quick to verify empirically, and it is unclear to us whether it would actually work in practice.

**Limitations:**

Yes.

**Strengths And Weaknesses:**

Strengths

1. A clean and well-motivated signal design. Only using the entropy over a single token as a proxy for reasoning sufficiency, and computing it is over 50× cheaper than rollout-based alternatives, as shown in Fig.6c.

2. The proxy model setup directly addresses the practical constraint that large API-deployed models do not have logits.

3. Careful and thorough experimental design. The paper covers multiple reasoning models, datasets, and baselines, and the appendix provides a solid set of ablation studies and analyses.

Weeknesses

EAT and Pass@1 correlation, which is the central assumption of the paper, currently rests entirely on empirical observation. The authors themselves acknowledge their limitation.

---

> ### Author Rebuttal · Authors · 2026-03-30
>
> We would like to thank the reviewer for the detailed and positive feedbacks.
>
> # Weakness: EAT and Pass@1 correlation
>
> Note that the Pass@1 (Avg@128) metric not only reflects the correctness, it also reflects the model's uncertainty in the answer:  If it converges to 1, implies that **all 128 rollout answers** are correct.
>
> So a converged Pass@1 --> model is absolutely certain in one answer.
>
> Our proposed entropy after `</think>` serves as a one-token proxy for the answer uncertainty and would therefore correlate with Pass@1 (Avg@128)
>
> # Question
>
> > 1. Could the authors clarify how EAT should be used in a realistic deployment setting where no such pre-filtering is possible?
>
> > 2. Is there a way to detect at inference time whether EAT is likely to be a reliable early-exit signal for a given problem?
>
> > 3. One potential direction would be to combine EAT with a complementary "stagnation" signal
>
> Thanks for raising these three very important questions.
>
> Now go back to the question:
>
> - For a realistic deployment setting, we need two conditions: An easy way to set the early stopping criteria, i.e. the variance threshold and no filtering. The first issue can be solved by using a separate validation set combined with finite sample correction, and the second issue can be solved by using a validation set + a complementary "stagnation" detection mechanism.
>
> - To detect whether EAT is reliable at inference time, we can look at EAT's behavior on the validation set to check whether early stopping with EAT can reach the user-specified risk tolerance and efficiency requirement.
>
> - To implement the stagnation detection, we can utilize a dynamic parametric EAT variance threshold that gradually decreases throughout reasoning (e.g. 1 - sigmoid((#token - shift) * slope)). When the actual EAT variance decrement goes over the threshold, we halt the reasoning and defer the instance to human experts, indicating that the model cannot solve this question. The tricky part here is to find the parameters of the parametric functions. Building on this work, [1] discusses the design of risk functions and demonstrates the feasibility of parametric threshold using a validaiton set under risk control framework for early stopping signals in general.
>
> [1] concurrent submission in supplementary material `submission_on_lower_threshold.pdf`

---

> > ### Author Rebuttal · Reviewer_qBAY · 2026-04-03
> >
> > Thank you for the author's detailed response, which has fully addressed my concerns. Although this work originates from a seemingly intuitive and modest insight, it delivers a thorough and rigorous analysis—encompassing a well-articulated motivation, meticulously designed methodology, comprehensive experiments, and a candid discussion of limitations. The paper is both intellectually rewarding and highly inspiring to read. Accordingly, I have decided to revise my score upward.

---

> > > ### Author Response · Authors · 2026-04-03
> > >
> > > We would like to thank the reviewer for their kind words and for acknowledging the value of our work.
> > >
> > > Per our discussion, we have included an additional section in the appendix providing a detailed guideline on the realistic deployment setting of EAT, including the choice of threshold, the extra threshold mechanism, the validation mechanism for the effectiveness of EAT.
> > >
> > > We believe this additional discussion can largely improve the practical impact of our work, making it more accessible to practitioners.
> > >
> > > Thanks,
> > >
> > > Authors

---

### Decision · Program_Chairs · 2026-04-30

**Decision:**

Reject

**Comment:**

This work argues that LLM overthinking methods are inefficient and proposes a simple, inexpensive, and novel signal, Entropy After </Think> (EAT), for monitoring reasoning and deciding whether to exit early. Experimental results show that EAT reduces token usage by 12–22% without harming accuracy. The main concerns are the novelty of entropy compared with prior entropy- or uncertainty-based approaches, and the central assumption that the entropy of the first token after </think> reflects reasoning convergence, which is not clearly justified. The rebuttal did not persuade the reviewers, and the work needs further improvement for acceptance.